# A comparison of lognormal and gamma size distributions for characterizing the stratospheric aerosol phase function from OPC measurements

Ernest Nyaku[1], Robert Loughman[1], Pawan K. Bhartia[2], Terry Deshler[3], Zhong Chen[4], and Peter R. Colarco[2]

[1]Center of Atmospheric Science, Hampton University, Hampton
[2]NASA Goddard Space Flight Center, Greenbelt, Maryland, 20771, USA
[3]Department of Atmospheric Science, University of Wyoming, Laramie, Wyoming
[4]Science Systems and Applications, Inc. (SSAI), 10210 Greenbelt Road, Suite 600, Lanham, Maryland 20706, USA

**Correspondence:** Ernest Nyaku (ernest.nyaku@hamptonu.edu)

**Abstract.**

A series of in situ measurements made by optical particle counters (OPC) at Laramie, Wyoming provides size-resolved stratospheric aerosol concentration data over the period 1971 - 2018. A subset of these data covering the period of 2008-2017 is analyzed in this study for the purpose of assessing the sensitivity of the stratospheric aerosol phase function to the aerosol size distribution (ASD) model used to fit the measurements. The two unimodal ASD models investigated are the uni-modal lognormal (UMLN) and gamma distribution models, with the minimum $\chi^2$ method employed to assess how well each ASD fits the measurements. The aerosol phase function ($P_a(\Theta)$) for each ASD is calculated using Mie theory, and is compared to the $P_a(\Theta)$ derived from the Community Aerosol and Radiation Model for Atmospheres (CARMA) sectional aerosol microphysics module. Comparing the $\chi^2$ values for the fits at altitudes of 20 and 25 km shows that the UMLN distribution better represents the OPC measurements; however, the gamma distribution fits the CARMA model results better than the UMLN model, when the CARMA model results are subsetted into the OPC measurement bins. Comparing phase functions derived from the UMLN distribution fit to OPC data with gamma distributions fit to CARMA model results at the location of the OPC measurements shows a satisfying agreement ($\pm 5\%$) within the scattering angle range of limb sounding satellites. This uncertainty is considerably larger if the CARMA data are fit with a UMLN.

## 1 Introduction

The presence of aerosol particles in the stratosphere has significant impact on atmospheric dynamics, atmospheric chemistry, and climate by altering the amount of radiation that reaches the Earth's surface, as research over the past few decades has shown (Kremser et al., 2016; Ivy et al., 2017). These aerosols form a layer of liquid droplets that are a mixture of sulfuric acid ($H_2SO_4$) and water ($H_2O$), discovered by Junge in 1960 (Junge et al., 1961). They can cool the Earth's surface and troposphere by scattering incoming short-wave radiation and warm the lower stratosphere by absorbing outgoing long-wave radiation (Robock, 2000; Kravitz et al., 2011; Ridley et al., 2014). These aerosols in the stratosphere act as condensation

nuclei for polar stratospheric clouds (PSCs), which provide a surface for heterogeneous chlorine activation and denitrification processes leading to ozone depletion (McCormick et al., 1995; Andreae and Crutzen, 1997; Solomon, 1999). Model simulations have shown that the Antarctic ozone hole was enhanced due to the addition of volcanic aerosols to the lower stratosphere that were associated with the eruption of Calbuco in 2015 (Ivy et al., 2017). The main sources of the stratospheric aerosols as summarized by Kremser et al. (2016) are from sulfur dioxide ($SO_2$) and carbonyl sulfide (OCS), which are both oxidized to sulfuric acid. OCS originates from marine sources and is transported by convection into the tropical stratosphere from the troposphere. Large volcanic eruptions directly inject plumes of $SO_2$ into the atmosphere that can reach beyond the tropical tropopause and into the stratosphere, where the $SO_2$ is oxidized and increases the load of the stratospheric aerosol particles (Solomon et al., 2011). This effect can last up to several years as was observed after the eruptions of El Chichón (Mexico, 1982) and Pinatubo (Philippines, 1991). The past 20 years have not experienced any large volcanic eruptions, but during this period the stratospheric aerosol load has been controlled by "moderate" but recurring volcanic eruptions that have been reported to be a primary source of the enhancement of global aerosol content (Vernier et al., 2011b; Neely III et al., 2013; Mills et al., 2016; Berthet et al., 2017). These moderate volcanic plumes can reach between 18-20 km in the lower stratophere, and through the upwelling branch of the Brewer-Dobson circulation, they are lofted into the mid-stratosphere up to 25 km altitude in about one year (Vernier et al., 2011b). Also, moderate volcanic eruptions like the Sarychev eruption in 2009 located in high-latitudes can enhance the aerosol load in the tropical stratosphere and even impact the other hemisphere (Wu et al., 2017). Other sources of stratospheric aerosols are from pyrocumulus clouds from large wild fires that can inject large amounts of combustion products and smoke into the stratosphere (Fromm et al., 2010; Khaykin et al., 2018) and from the transport of $SO_2$ from the surface to deep into the stratosphere through the Asian monsoon whose circulation provides an effective avenue for pollution from Asia, Indonesia, and India to enter the global stratosphere (Randel et al., 2010; Vernier et al., 2011a; Ploeger et al., 2017; Yu et al., 2017).

The stratospheric aerosol phase function $P_a(\Theta)$ describes the angular distribution of the scattered solar radiation and it depends on the size, shape, and refractive index of the aerosol. The value of the phase function for a given scattering angle is proportional to the probability that an incident photon will be scattered in a particular direction. Theoretically, the $P_a(\Theta)$ is calculated from the aerosol size distribution (ASD) using Mie theory (Mie, 1908), generally assuming that the aerosol particles in the stratosphere are spherical and homogeneous. In this study, a refractive index of $1.45 + 0i$ is assumed as appropriate for hydrated sulfuric acid (Palmer and Williams, 1975) and is used for all wavelenghs. An estimate of the actual $P_a(\Theta)$ is needed to interpret limb scatter (LS) measurements (Rault and Loughman, 2007; von Savigny et al., 2015), and LIDAR measurements, in order to estimate the aerosol extinction profile needed for the aerosol forcing calculations. The $P_a(\Theta)$ estimate is not needed for satellite measurements which use occultation, which allows extinction to be derived directly. The actual ASD varies in space (latitude, longitude, and altitude) and time, but scattering based retrievals rarely include this variation.

The Ozone Mapping and Profiler Suite, Limb Profiler (OMPS/LP) (Flynn et al., 2006; Rault and Loughman, 2013; Jaross et al., 2014), the Optical Spectrograph and InfraRed Imaging System (OSIRIS) (Llewellyn et al., 2004) and Scanning Imaging Absorption spectroMeter for Atmospheric CartograpHY (SCIAMACHY) (Bovensmann et al., 1999) are three limb scattering instruments that have been mounted on satellite platforms to measure limb scattered sunlight. These satellite instruments have

measured the limb radiance profiles from wavelengths ranging from the UV to the near infrared from which stratospheric aerosol extinction profiles, the standard operational product (for OMPS/LP and OSIRIS) are retrieved. The retrieval of stratospheric aerosol extinction profiles from limb radiance measurements (Rault and Loughman, 2007; Bourassa et al., 2007, 2008; Taha et al., 2011; Ovigneur et al., 2011; Bourassa et al., 2012; Ernst, 2013; von Savigny et al., 2015; Rieger et al., 2015, 2018; Loughman et al., 2018), involves the comparison of measured limb radiance data with simulated radiances that are generated by radiative transfer (RT) models. This approach has also been used to obtain the ASD information of stratospheric aerosol from limb scatter measurements (Malinina et al., 2018).

Several ASDs that are used in the aerosol extinction retrieval algorithms by the various LS instruments are presented in Table 2 of Loughman et al. (2018). In many cases, the assumed size distributions used for the derivation of the phase functions are based on lognormal distribution fits made to the University of Wyoming balloon-borne optical particle counter (OPC) measurements that have been made at different places and times. These fits were made prior to the OPC corrections proposed by Kovilakam and Deshler (2015) and Deshler et al. (2019). The $P_a(\Theta)$s derived from the OPC measurements are used in the computation of the limb radiances, which are then compared to the measured radiances to retrieve the aerosol extinction coefficients. As a result, the retrieved aerosol extinction is related to the $P_a(\Theta)$ employed in the retrieval process. Mie theory shows that the shape of the $P_a(\Theta)$ varies considerably with particle size and refractive index. Thus for spherical sulfuric acid droplets in the stratosphere with a known refractive index, as the particle size increases, the shape of the $P_a(\Theta)$ changes from a simple Rayleigh symmetric phase function to a more complex one with more forward scattering (Boucher, 1998).

A long historical record of stratospheric aerosol monitoring is available from the Stratospheric Aerosol and Gas Experiment (SAGE) solar occultation data. This measurement technique was begun by the Stratospheric Aerosol Measurement (SAM) and then the SAGE series provided a nearly continuous data record from 1984 -2005 (Russell and McCormick, 1989; McCormick and Veiga, 1992; Thomason et al., 1997). SAGE solar occultation data provides a weak constraint on the ASD through the wavelength dependence of the retrieved aerosol extinction, but cannot be used to uniquely retrieve the ASD (Yue, 1999; Thomason et al., 2008). In spite of the limited information in the SAGE solar occulatation data, Bingen et al. (2004) made an attempt to derive a global climatology of stratospheric aerosol size distribution parameters from SAGE II extinction profiles by assuming a lognormal ASD. Since the $P_a(\Theta)$ used in the radiative transfer models are computed from assumed ASDs, different groups have used various techniques to model it. Some techniques that have been used to model the $P_a(\Theta)$ are by computing it using the Henyey-Greenstein phase function (H-G) (Henyey and Greenstein, 1941; Ernst, 2013; Grams, 1981) or the modified Henyey-Greenstein phase function (MH-G) (Irvine, 1965; Cornette and Shanks, 1992) with a precise asymmetry factor $g$, which is the average cosine of the scattering angle weighted by the phase function. The shortcomings of using these functions to approximate the real Mie phase function were demonstrated by Toublanc (1996) for two cases. When the radius of the particle was ten times smaller than the wavelength, the H-G phase function failed to produce the shape of the real Mie phase function in comparison to that of the MH-G. By contrast, for a particle of radius that was comparable to the wavelength, both functions failed to reproduce the lobe patterns of the real Mie phase function.

The OSIRIS version 5 and 7 (Bourassa et al., 2012; Rieger et al., 2019), the OMPS version 0.5 (Loughman et al., 2015; DeLand et al., 2016) and the SCIAMACHY versions 1.1 and 1.4 (von Savigny et al., 2015; Rieger et al., 2018) aerosol

extinction retrievals use a single-mode lognormal ASD to model $P_a(\Theta)$, by using the median radii ($r_m$) and widths ($\sigma$) given in Table 1. For both algorithms, $P_a(\Theta)$ does not vary with altitude or location. The recently developed V1 (Loughman et al., 2018) and V1.5 (Chen et al., 2018) OMPS aerosol extinction retrievals use updated bi-modal lognormal and gamma phase functions respectively. The Ångström exponent (AE) (Ångström, 1929) is a parameter that captures the variation of the aerosol extinction with wavelength, which provides some indication of particle size.The AE is computed using Equation (1), where $K_{ext}$ is the aerosol extinction coefficient derived using the ASD and Mie theory for the two wavelengths of interest (525 nm and 1020 nm).

$$AE = \frac{-ln[K_{ext}(\lambda_1)/K_{ext}(\lambda_2)]}{ln[\lambda_1/\lambda_2]} \tag{1}$$

**Table 1.** Size distribution parameters used to calculate the aerosol phase functions of the various versions of OMPS, OSIRIS v5, and SCIAMACHY. The Ångström exponent (AE) is derived from Equation (1) using the 525 nm and 1020 nm extinction coefficients.

| Instrument (Data Version) | Distribution | CMF | $r_{m_i}(\mu m)$ | $\sigma_i$ | $AE$ |
|---|---|---|---|---|---|
| OMPS (V0.5) | UMLN | - | 0.06 | 1.73 | 2.34 |
| OMPS (V1.0) | BMLN | 0.003 | 0.09, 0.32 | 1.4, 1.6 | 2.01 |
| OMPS (V1.5) | Gamma | - | $\alpha$ =1.8 | $\beta$ =20.5 | 2.078 |
| OSIRIS (V5) / SCIAMACHY (V1.4) | UMLN | - | 0.08 | 1.60 | 2.44 |
| SCIAMACHY (V1.1) | UMLN | - | 0.11 | 1.37 | 2.82 |

Comparison of the extinction coefficients derived from OPC measurements and SAGE II occultation have shown differences that vary by more than 50% particularly for non-volcanic periods (Kovilakam and Deshler, 2015); however, these differences have been largely eliminated after the calibration error identified by Kovilakam and Deshler (2015) was accounted for in the new method to derive uni/bi-modal lognormal size distributions to fit OPC measurements (Deshler et al., 2019). With the application of this new size distribution retrieval method the extinctions estimated from the OPC measurements agree, within the measurement uncertainty, with both SAGE II and HALOE extinctions nearly throughout the altitude and time periods of these measurements.

The aerosol signal in the measured LS radiance at a given tangent height is proportional to the product of the aerosol extinction in that layer and the aerosol phase function at the tangent point provided the path is optically thin. Rieger et al. (2018) have shown that when the radiance is simulated to include coarse mode particles in the atmosphere with an assumed AE, then the differences between the lognormal parameters used in the simulation and the retrieval induces errors in the retrieved aerosol extinction as function of AE which corresponds to 30% for OSIRIS geometries and 50% for SCIAMACHY geometries. This is because the phase functions of BMLN distributions vary more widely for a given AE and this leads to a complicated relationship with the retrieved error. The analysis of Rieger et al. (2018) illustrates the importance of the assumed

value of $P_a(\Theta)$ for LS retrievals of $K_{ext}$, and the limitations of using AE alone to estimate the value of $P_a(\Theta)$, whereas the analysis of Deshler et al. (2019) illustrates that closure is achieved between well characterized in situ measurements and solar occultation extinction measurements.

This paper seeks to first show the differences that arise in the computed $P_a(\Theta)$ when different size distribution functions are fitted to the same aerosol concentration measurements. The sensitivity of the derived $P_a(\Theta)$ value to the presence or absence of aerosol concentration information in the aerosol size range of 0.01 μm and 0.1 μm is also explored. Section 2 briefly describes some of the ASDs that have been used in the past to characterize the stratospheric aerosol load and a description of the ASD that is derived from aerosol concentration based on data from Laramie, Wyoming optical particle counter (OPC) measurements using balloon-borne instruments (Deshler et al., 2003; Ward et al., 2014; Deshler et al., 2019). Section 3 focuses on a study which is based on the 2008 - 2017 OPC data at the same location by comparing the unimodal lognormal (UMLN) and the gamma distribution fits to this data set. This is done by concentrating on two altitudes 20 km and 25 km and noting the differences between the phase functions and the Ångström exponents of these distributions. The two distributions are then compared to the outputs of the Community Aerosol and Radiation Model for Atmospheres (CARMA) sectional aerosol microphysics module running online in the NASA Goddard Earth Observing System (GEOS) model. We conclude with a summary and recommendations on which distribution to choose depending on what kind of stratospheric aerosol measurements are available.

## 2   Aerosol Size Distribution

The aerosol size distribution or the particle number density per unit radius is a statistical model used to describe an ensemble of particles. A number of particle distributions such as the Junge power-law (Junge, 1963), the modified gamma (Deirmendjian, 1969) and up to seven lognormal (Davies, 1974) distributions have been used in the past to represent the distribution of aerosols in the atmosphere. A comprehensive description and a comparative presentation of these distributions is given by Deepak and Box (1982) or Hinds (1982).

For the characterization of aerosols in the stratosphere, lognormal (LN) size distributions are commonly used, although other distributions have been tried in the past (Toon and Pollack, 1976; Rosen and Hofmann, 1986; SPARC, 2006). A discussion of fitting LN distributions to the aerosol measurements obtained from OPC is given by Horvath et al. (1990). LN aerosol size distribution parameters for stratospheric aerosols have also been been retrieved from LS measurements (Rault and Loughman, 2013; Rieger et al., 2014; Malinina et al., 2018). The UMLN distribution consists of three parameters: The total aerosol concentration and two parameters that indicate the median radius and width of the ASD. The bimodal lognormal (BMLN) distribution became the favored function for fitting stratospheric aerosol concentration measurements since the eruption of Mount Pinatubo injected large quantities of sulfur dioxide ($SO_2$) into the stratosphere (Deshler et al., 2003). A multimodal distribution can be used to represent coexisting "nucleation" , "coagulation", and "accumulation" modes after a volcanic eruption. The nucleation mode is associated with new particle formation from sulfuric acid vapor which quickly coagulate to form larger particles

(Hamill et al., 1997), and the accumulation mode associated with particle growth by condensation of the vapor on the existing particles (Steele and Turco, 1997).

## 2.1 ASD from Wyoming OPC measurements

Stratospheric aerosol measurements to altitudes above 30 km have been taken from balloon-borne platforms at Laramie, Wyoming since 1971 with a one liter per minute two channel OPC originally developed by Rosen (1964) and then with a modified 10 liter per minute 8 -12 channel counter (Hofmann and Deshler, 1991). The instrument measures the intensity of scattered white light at 25° (Rosen, 1964) and 40° (Hofmann and Deshler, 1991) in the forward direction from single particles passing through the light beam, which is larger than the air sample stream. See Table 1 of Deshler et al. (2003) for the measurement history up to 2003. The instrument calibration and the instrument response function which depends on the light source, the detector efficiency, and Mie theory are used to determine aerosol size from the intensity of the scattered light. The size resolved OPC number concentration measurements are then fitted with an assumed functional form for the size distribution to describe the measurements. The measured concentrations are fitted by either a UMLN or a BMLN size distribution at each measured altitude, where the particle concentrations at distinct size bins are fitted with the function defined by Equation (2) (Deshler et al., 2019), where the sum is over either $n = 1$ or 2 modes. Prior to the analysis of Deshler et al. (2019), Equation (2) was used without including the channel dependent counting efficiency function (CEF$_{ch}$) (Deshler et al., 1993, 2003).

$$N_{ch} = \int\limits_{0}^{\infty} \left[ \sum_{i=1}^{n} \frac{N_i}{\sqrt{2\pi} ln[\sigma_{mi}]} \exp\left( \frac{-ln^2[x/r_{mi}]}{2ln^2[\sigma_{mi}]} \right) \right] CEF_{ch}(x) d\, ln(x). \tag{2}$$

This distribution assumes that the measured concentrations are normally distributed with respect to the logarithm of the radius for each mode of the distribution. While the OPCs in use since 1991 employ 8 to 12 aerosol channels, the number of measurements decrease with increasing radius channels and altitude as the concentration of the larger particles decrease below detection thresholds. A minimum of four size resolved concentration measurements are required to fit a bimodal distribution. The fifth measurement is obtained from the measurement of the total aerosol population using a condensation nucleus counter (Campbell and Deshler, 2014). The sixth measurement is obtained from the first channel with no aerosol counts, providing an upper limit on the aerosol concentration at that size. Thus, for every mode of the lognormal distribution, $N_i$ represents the total number concentration, $r_{mi}$ is the median radius, $\sigma_{mi}$ is the mode width. The best fit is the distribution (BMLN or UMLN) which minimizes the sum over all measured sizes of the root mean square difference of the logarithm of the fitted concentration and the log of the measured concentration. This method of searching for the best fitting parameters is quite similar to the chi-square technique described below, where here the use of logarithms provides the normalization by particle number concentration. Measurement uncertainties arise from variations of air sample flow rate, Poisson counting statistics, and the ability to duplicate the measurements from two identical instruments. The impact of these uncertainties on the size parameters have been approximated by a Monte Carlo simulation to be ±30% for size distribution parameters and ±40% for the aerosol moments (Deshler et al., 2003). A systematic calibration error affecting the counting efficiency of the instruments

was described by Kovilakam and Deshler (2015). The discovery of this error has led to a modification in the fitting algorithm described in Deshler et al. (2003), such that now an explicit counting efficiency is included in the derivation of the lognormal size distribution fitting parameters (Deshler et al., 2019), as indicated by CEF$_{ch}$ in Equation (2).

During background stratospheric aerosol conditions, OPC measurements may not provide sufficient information about smaller particles ($r < 0.15$ μm) to determine a robust BMLN fit as shown in a recent study to improve OMPS/LP aerosol retrievals by Chen et al. (2018). In that study, Chen et al. (2018) compared four BMLN fits to the same OPC data at 20km altitude (made on 12 April 2000), all having a similar AE of approximately 2.4, but each with a different coarse mode fraction (CMF). These four BMLN distribution fits to the OPC data differed significantly from each other in the radius range between 0.01μm to 0.1μm (see Figure A1 of Chen et al., 2018), a region of the size distribution which is very challenging to measure with an OPC due to the small amount of light scattered by such small particles. These physical limitations on OPC measurements in turn limit the ability of the fits to be constrained. Consequently, the different ASDs produced $P_a(\Theta)$ that differed significantly from each other in backscatter as shown in Figure A2 of Chen et al. (2018). Additionally, the BMLN distribution, which is defined by 5 parameters (the CMF, 2 median radii, and 2 mode widths) that are independent of each other at each altitude, cannot generally be determined in cases where the measurements have less than 5 data points. This limitation is further explored in the next section through a reexamination of the OPC data by fitting 2 single mode distributions to the concentration measurements and using only data available since 2008 that has a measurement between the 0.05 μm and 0.1 μm range.

## 3 Reanalysis of OPC size distribution fits

For a reanalysis of the OPC measurements, it was assumed the stratospheric aerosol could be described by a unimodal distribution during the non-volcanic period under consideration (2008-2017). Either a lognormal or a gamma model is used for which the number of degrees of freedom is reduced from five to two for a normalized distribution during the fitting process (described in Section 3.1). The normalization of the concentrations has no effect on the computation of the $P_a(\Theta)$ and the Ångström exponent for this study. The goodness of the fits is determined by the minimization of the chi-square ($\chi^2$) test statistic. This method estimates the parameters of the fitted distribution by minimizing the difference between the hypothesized and observed distributions. If the data are grouped into $k$ categories ($i = 1, 2, 3, ..., k$) according to the magnitudes of the radii, the observed frequency in each class is denoted by $O_i$, and the expected probability from the hypothesized distribution by $\zeta_i$, then the $\chi^2$ value can be calculated from Equations (3) and (4), corresponding to Equation (5.14) described by Wilks (2011)

$$\chi^2(\xi) = \sum_{i=1}^{k} \frac{[\frac{O_i}{n} - \zeta_i(\xi)]^2}{\zeta_i(\xi)} = \sum_{i=1}^{k} \frac{[O_i - n\zeta_i(\xi)]^2}{n^2\zeta_i(\xi)} \tag{3}$$

and

$$n = \sum_{i=1}^{k} O_i. \tag{4}$$

To minimize the $\chi^2$ value, the parameters ($\xi$) of the hypothesized distribution $\zeta$ are adjusted until the $\chi^2$ value closest to zero is obtained (Cho et al., 2004). Thus, if the fitted distribution is closer to the distribution of the data, the expected number of particles and the observed number of the particles are very close for each radii range, and the square of the differences in the numerator of Equation (3) would be very small, leading to a small $\chi^2$ (Wilks, 2011).

5    To assess whether the number of particles is being over or under estimated between the 0.05 μm and 0.1 μm radii range, data that includes a measurement within this range should be used, but such measurements are not generally available due to inherent limitations on the sensitivity of generic OPCs to particles less than 0.1 μm. Generally, the Wyoming in-situ OPC aerosol concentration measurements include size resolved concentrations for particles between 0.15 μm and 2.0 μm in 12 size classes. In addition, a second instrument is used to provide the concentration of all particles > 0.01 μm using a condensation 10  nuclei counter which provides no size information. Beginning in 2008 the OPC developed in the late 1980s (Hofmann and Deshler, 1991) was replaced with a new laser based OPC, or LPC (Ward et al., 2014), which is sensitive to particles from 0.092 to 4.5 μm radius in 8 size classes. On certain occasions, between 2008 and 2010, there were measurements from both the older OPC and the newer LPC deployed on the same balloon. Further analysis and discussions will explore the importance of the additional bin at a radius of 0.092 μm by comparing the fits that include this bin to those that were fitted excluding this bin, as 15  well as the resulting phase functions derived from the fits in section 3.2.

### 3.1   Unimodal lognormal or gamma distribution

Aerosol concentration measurements from Laramie, Wyoming with the LPC are used for the current study because of the inclusion of a measurement between 0.05 μm and 0.1 μm. The LPC data consists of 27 months of measurements as shown in Table 2 made from 2008 to 2017 that are fitted with the cumulative forms of the normalized UMLN distribution and the gamma 20  distribution. These data are available from University of Wyoming (2018). The normalized form of the cumulative UMLN is obtained by setting $N_i$ of Equation (2) to one and the cumulative gamma distribution is given by Equation (5).

$$F(x,\alpha,\beta) = \int_0^x f(u;\alpha,\beta)du \quad = \frac{\gamma(\alpha,\beta,x)}{\Gamma(\alpha)} \tag{5}$$

In the case of the cumulative gamma distribution, $\gamma(\alpha,\beta,x)$ is the lower or incomplete gamma function, where $\alpha$ is the shape parameter and $\beta$ is the rate parameter. The mean and variance of this distribution are respectively given by $\alpha\beta$ and $\alpha\beta^2$. 25  This distribution can display many shapes by altering the values of $\alpha$ and $\beta$, and the pliable shape of this distribution makes it a good candidate for representing stratospheric aerosol data. A difficulty with this distribution, as stated by Wilks (2011), is that it is more tedious to work with the gamma distribution because the two parameters do not correspond exactly to the physical parameters of the size distribution of the sampled sampled data, as is the case for the lognormal distribution.

**Table 2.** Table showing the year and the months on which the LPC data was included in this study. Each month represents one LPC flight with stratospheric measurements.

| Year | Month |
|------|-------|
| 2008 | October |
| 2009 | January   June   November |
| 2010 | March   June |
| 2011 | March   June   July   November |
| 2012 | January   March   May   July   September   November |
| 2013 | March   May   August   October |
| 2014 | March   July   September   November |
| 2015 | January |
| 2016 | April |
| 2017 | November |

The two altitudes, 20 km and 25 km are chosen to represent two differing aerosol loads well away from the tropopause. For the new fits, the measurements for each aerosol radius bin size which are reported as cumulative number concentrations ($N_i$) are first normalized to the total aerosol concentration $N_0$. This value represents the total number concentration and is obtained at the lowest integration limit of 0.01μm. After the normalization, bins that have quantities less than $1 \times 10^{-6}$ cm$^{-3}$ are omitted

because this number is less than the smallest count distinguishable by the instrument used to make the measurements, which is $\sim 10^{-5}$ cm$^{-3}$ (Deshler et al., 2003). The best fit (for which the $\chi^2$ is minimized) is then chosen as the fit for that particular distribution. Examples of the fitted cumulative UMLN and the gamma distributions to the OPC data for the two altitudes are shown in Figure 1 for the June 2010 data. The two ASDs tend to diverge beginning at radii greater than 300 nm and differ substantially at approximately 600 nm, where aerosol concentrations are below the minimum detectable concentrations, and

there these differences can reach one order of magnitude. The figures also display for each fit the AE that was computed using Equation (1), where $\lambda_1$ and $\lambda_2$ are 525 nm and 1020 nm respectively.

To determine which of the two distributions was a better fit to the available data of each month's measurements, a statistical significance test was conducted by using the $\chi^2$ goodness of fit test. This was done such that the null hypothesis $H_o$ stated that: for each measurement the data were drawn from either a UMLN or a gamma distribution. The $\chi^2$ is used as the test statistic

with the degrees of freedom $v$ given by Equation (6).

$$v = Number\ of\ measured\ bins - 2 - 1 \tag{6}$$

The number 2 in this equation represents the two parameters ($r_m$ and $\sigma$, $\alpha$ and $\beta$, for the UMLN or the gamma, respectively) that are fitted for each distribution. The percentile value is defined as the distinct probability that the observed value of the

test statistic will occur according to the null hypothesis. Subsequently, the null hypothesis is rejected if the percentile value is less than or equal to the test level and it is not rejected otherwise (Wilks, 2011). A complete summary of the percentile values computed for each of the two altitudes for all the data considered in comparing the two distributions is given in Figure 2. Results from this figure indicate that at both altitudes 20 km and 25 km, the null hypothesis is not rejected at the 15% test

level (this corresponds to percentile values greater than 0.85). This signifies that at this level of significance the data could have been drawn from either a UMLN or gamma distribution. But at a 5% test level which corresponds to a percentile value greater than 0.95, the null hypothesis is rejected in favor of the alternate hypothesis that the data are taken from a UMLN distribution throughout the record at 25 km and for a majority of measurements at 20 km. Thus, the UMLN distribution is the better of the two distributions that were fitted to the data for the two altitudes that were used in this study.

The fitted parameters are then used to derive the phase functions which are compared among the two distributions. The phase functions derived from the parameters of both distributions compare well to within 10% of each other for scattering angles greater than 20°. Example of the derived phase functions for 675 nm (wavelength used to perform aerosol extinction retrieval by OMPS V1.0 ) using the UMLN and the gamma distributions fitted parameters displayed in Figure 1 are shown in Figure 3. The shape of the phase functions has been observed to depend primarily on magnitude of the median radius ($r_m$) in

the case of the UMLN distribution and the shape parameter ($\alpha$) in the case of the gamma distribution. As the magnitude of these two parameters decreases, indicating smaller particles, the resulting phase functions produced from either of the distributions would more closely resemble a Rayleigh phase function, as is suggested in Figure 3, where the 25 km distribution (which has smaller particles) is compared to the 20 km, distribution (which has larger particles and hence larger values of $r_m$ and $\alpha$). Phase functions derived from the same dataset but using different fitting models differ from each other and this is a very

important issue in the interpretation of measurements from scattering instruments. This is further compounded especially for limb scattering instruments due to multiple scattering effects, since differences in the phase functions produce reflectivity and altitude dependent differences in derived extinctions (Chen et al., 2018). Figure 3 also shows the range of scattering angles observed by OMPS-LP (Loughman et al., 2018), SCIAMACHY (von Savigny et al., 2015) and OSIRIS (Rieger et al., 2018) limb scattering satellite instruments.

The AE computed from the parameters of UMLN distribution fits to data are similar to those computed from the gamma distribution fitted parameters for the same altitude, as is shown in Figure 4. Moreover, a large AE corresponds to a small median radius in the case of the UMLN distribution or to a small shape parameter in the case of the gamma distribution.

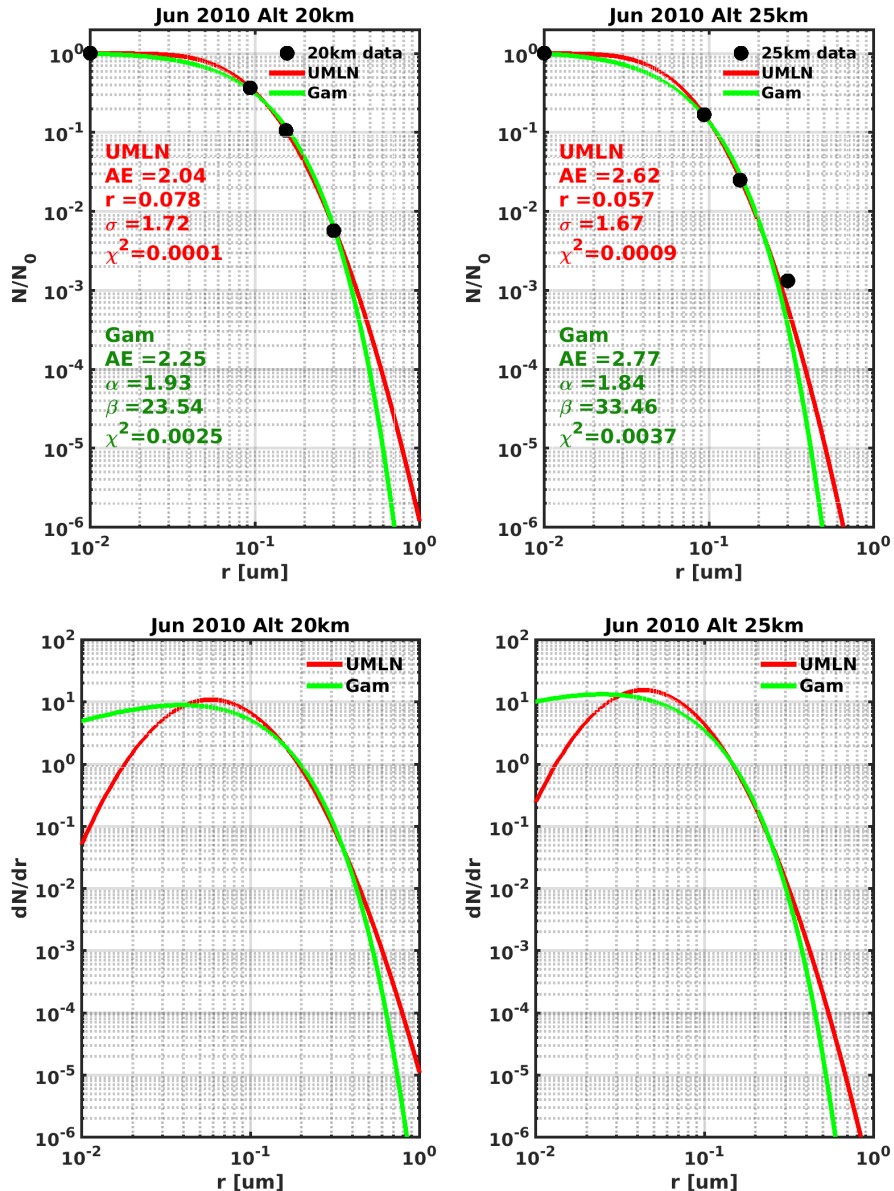

**Figure 1.** Topmost figures show examples of fitting the cumulative form of the UMLN (red line) and the gamma (green line) distributions to the June 2010 OPC data at altitudes 20 km (left) and 25 km (right) using the minimum $\chi^2$ technique. The figures also show the results of each fit, the minimized $\chi^2$ and the Ångström exponent derived from the fitted parameters. The figures at the bottom show the differential form of the two distributions.

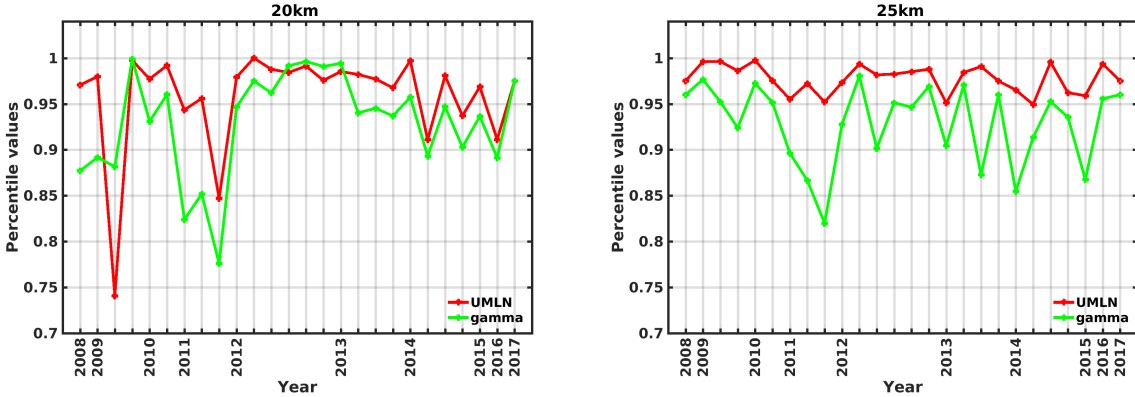

**Figure 2.** Percentile values computed for the $\chi^2$ values of the UMLN (red line) and gamma (green line) distribution fits to 2008 to 2017 OPC data for altitudes 20 km (left) and 25 km (right).

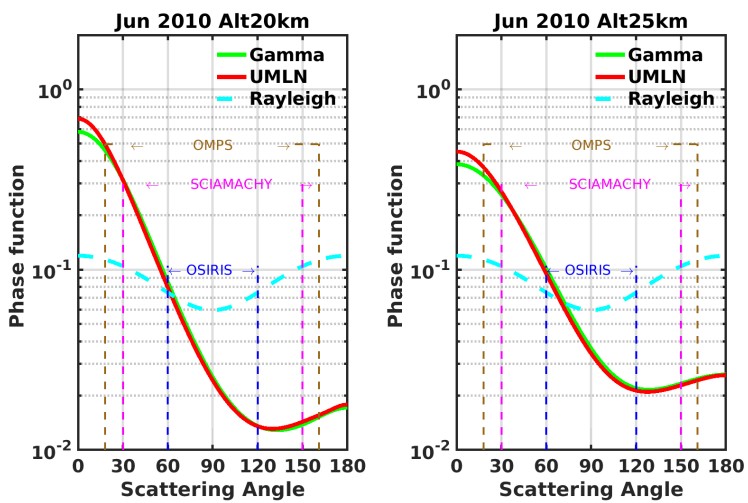

**Figure 3.** Phase functions derived at 675 nm using the fits shown in Figure 1 for June 2010. The figures correspond to altitude 20 km (left) and to altitude 25 km (right). Also shown is the range of scattering angles for which aerosol extinctions are retrieved for OMPS, SCIAMACHY and OSIRIS limb scatter instruments.

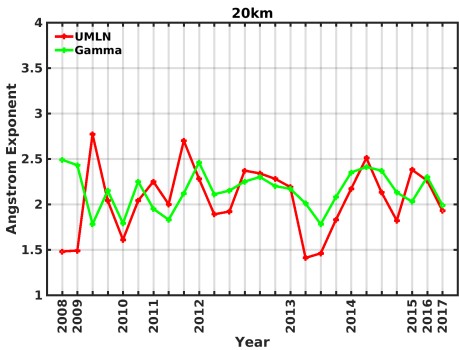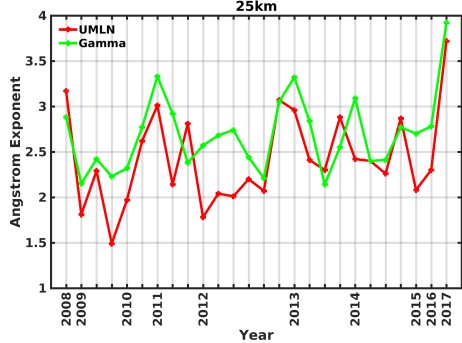

**Figure 4.** Computed Ångström exponents for both the UMLN (red lines) and the gamma (green lines) distribution at altitude 20 km(left) and 25 km (right).

## 3.2 Importance of a measurement between 0.05 μm and 0.1 μm

The form of $P_a(\Theta)$ for a particular aerosol is determined by the value of the size parameter $X$, which is the ratio of the aerosol circumference to the wavelength of interest ($X = \frac{2\pi r}{\lambda}$). Examples of phase functions for mono disperse aerosols for different $X$ are shown in Figure 5. From this figure, the greatest sensitivity for the forward scattering angles of $P_a(\Theta)$ occurs when

5    $X = 3$ and this implies an aerosol radius $r \approx 0.3$ μm for a wavelength of 675 nm. The phase function for $X = 3$ shows a forward peak and is nearly constant for scattering angles $\Theta \geq 70°$. When there are no measurements between the 0.01 and 0.15 μm bin sizes, then the particle concentration within this range is estimated by the function used to fit the data. Errors in estimating the number of particles within this range by the function used for fitting the data may lead to uncertainties in the phase function; however the contribution of particles within this range to the overall scattering is small due to the strong size

10    dependence of the scattering cross section. Compare the phase function plots shown by the $X = 1$ and $X = 3$ lines in Figure 5. For an aerosol radius $r = 0.1$ μm, $P_a(\Theta)$ is approximately $30\%$ greater than the Rayleigh phase function for scattering angles less than $60°$. The additional 0.092 μm bin in the LPC will augment the measurements.

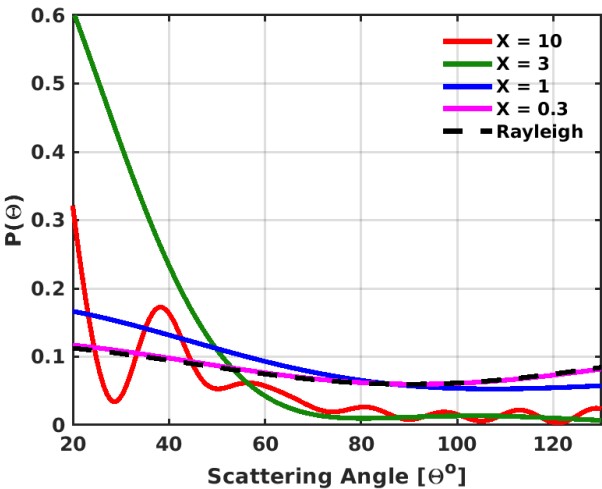

**Figure 5.** Mie phase functions of a monodisperse aerosol for different values of the size parameter $X$ derived with a refractive index of 1.45+0i. The increasing asymmetry and complexity (e.g. for $X$=10) of the phase functions with increasing $X$ is due to the use of a monodisperse aerosol. The oscillations observed are damped when the phase functions are computed for an ensemble of aerosols that are assumed to have a UMLN or gamma distribution. The phase functions are shown for the range of scattering angles that are observed by OMPS, SCIAMACHY and OSIRIS.

The fits shown in Figure 1 are repeated for each of the two distributions, this time excluding the 0.092 µm bin. These no small bin (nsb) fits are called UMLN$_{nsb}$ and gamma$_{nsb}$, and the resulting $P_a(\Theta)$ are compared at each altitude. A typical fit showing how the two distributions performed is shown in Figure 6 for June 2010 data at altitude 25 km. The topmost panels shown in Figure 6 indicate that the UMLN$_{nsb}$ distribution tends to underestimate the measured concentration at the 0.092 µm bin

position, whereas the same behavior is not seen with gamma$_{nsb}$. These panels also show that only the UMLN$_{nsb}$ does a good job of fitting the 0.3 µm point, while both gamma distributions miss this point similar to UMLN. Also included in this figure are the $P_a(\Theta)$ (middle plots) determined for the 675 nm wavelength from the fitted parameters of both distributions. The range of scattering angles for which aerosol extinction retrievals are performed by OMPS, SCIAMACY and OSIRIS are indicated in this figure. The corresponding $P_a(\Theta)$ ratios comparing the different fits are shown in the bottom plots. Changes of up to $\pm30\%$

depending on the scattering angle are seen between the derived UMLN distribution $P_a(\Theta)$ comparison (UMLN/UMLN$_{nsb}$), whereas changes of up $\pm10\%$ are observed for the derived gamma distribution $P_a(\Theta)$ comparison (Gam/Gam$_{nsb}$). The large differences between the UMLN $P_a(\Theta)$ and both UMLN$_{nsb}$ and gamma$_{nsb}$ is mainly due to the difference between the fits at 0.3 µm rather than particle radii less than 0.1 µm. Thus, underestimating the 0.3 µm data point leads to a reduction in the $P_a(\Theta)$ for the forward scattering angles and an increase in the phase function for the backward scattering angles. Also, since

both gamma distributions underestimate the 0.3 µm point and are otherwise quite similar, their phase functions show very little variation due to the differences between the fits at 0.05 µm and 0.1 µm. Additionally, both UMLN and gamma distribution fits underestimated the particle radius at approximately 0.3 µm and a comparison of their phase functions (Gam/UMLN)

show variations of $\pm 10\%$ for scattering angles greater than 30°. The failure of both gamma distributions to capture the OPC measurements for the largest bin size for the case shown in Figure 6 could lead to a systematic error in the derived phase functions.

The conclusion drawn from this comparison is that the phase functions calculated with the gamma distributions with and without the small bin are comparable to each other to within 10% as compared to those of the UMLN distribution. This signifies that the gamma distribution is relatively insensitive to the addition of an intermediary bin between 0.05 μm and 0.1 μm, whereas the UMLN distribution is quite sensitive to this additional information.

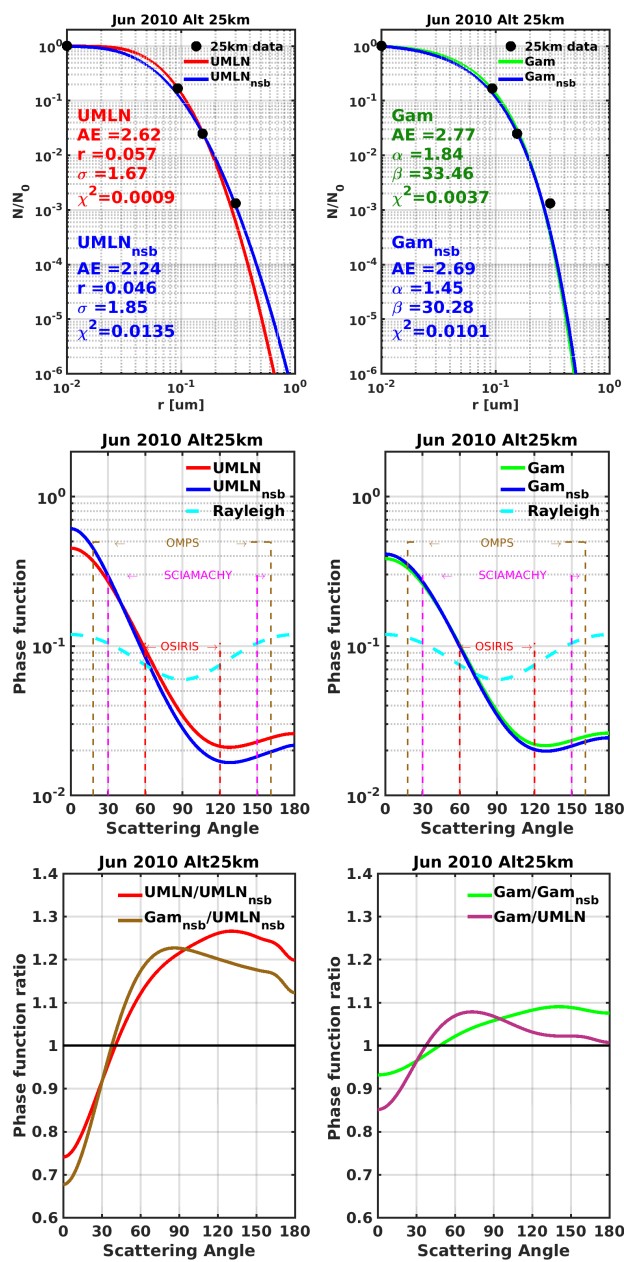

**Figure 6.** Unimodal lognormal distribution fits (top left) and gamma distribution (top right) fits to June 2010 data for altitude 25 km. Blue lines indicate fits$_{nsb}$ made without the 0.094 μm measurement, while the red line fit includes all measurements as before (compare with Figure 1). The middle figures are the phase functions derived at 675 nm wavelength from the parameters of the fits. The range of scattering angles observed by OMPS, SCIAMACHY and OSIRIS for which aerosol extinction retrievals are performed are also indicated on these figures. The bottom figures show the ratios of the phase functions of the different fits.

### 3.3 Comparison to the CARMA microphysical model results at Wyoming

The Community Aerosol and Radiation Model for Atmospheres (CARMA) is a general-purpose sectional microphysics code, which was derived from a one-dimensional stratospheric aerosol code that was developed by Turco et al. (1979); Toon et al. (1979, 1988) to study aerosols and clouds in planetary atmospheres (Hartwick and Toon, 2017). This model includes both aerosol microphysics and gas phase sulfur chemistry that has been described by English et al. (2011). CARMA has been implemented in the NASA Goddard Earth Observing System (GEOS) Earth system model (Rienecker et al., 2008; Colarco et al., 2014) and configured for modeling stratospheric aerosols similar to English et al. (2011). The ASD is not defined by a statistical distribution but instead is handled using a number of discrete size bins, where the model transport processes are allowed to affect each size bin independently (Colarco et al., 2014). The $P_a(\Theta)$ produced by this model is computed directly using the outcome of each discrete bin to perform Mie calculations. The current configuration of this model employs 22 size bins ranging from 0.0002 μm to 2.79 μm at 72 vertical levels from the surface of the Earth up to 85 km. The distribution of the aerosol size bins is shown in Table 3. The model output for this comparison is the June-July-August (JJA) climatology that was averaged over 2008 to 2017 at Laramie, Wyoming. During this period the eruptions of Kasatochi, Sarychev Peak, and Calbuco did not influence the atmosphere above 20 km over Laramie. Thus during this period, the atmosphere contains the background stratospheric aerosol layer, precursor emissions for anthropogenic sulfates, and degassing volcanoes that are not explosive in nature. The evolution of particles for this model arises from the nucleation of new sulfate particles, condensation of sulfuric acid vapor onto existing sulfate particles, and subsequent coagulation of sulfate particles. The cumulative UMLN size distribution and the cumulative gamma distribution are then fitted to the model results at altitudes between 19 and 26 km using Equations (2) and (5) respectively.

**Table 3.** Table shows the distribution of the 22 aerosol size bins for the radii of the particles employed in the CARMA model.

| 0.000267 μm | 0.0004 μm | 0.0006 μm | 0.001 μm | 0.0016 μm | 0.002 μm | 0.0038 μm | 0.0058 μm |
|---|---|---|---|---|---|---|---|
| 0.009 μm | 0.014 μm | 0.022 μm | 0.034 μm | 0.053 μm | 0.082 μm | 0.128 μm | 0.198 μm |
| 0.308 μm | 0.479 μm | 0.744 μm | 1.156μm | 1.796μm | 2.79 μm | | |

The cumulative distribution fits are performed according to the methodology described in section 3.1 and using selected radii bins in conformity to the size resolved OPC measurements. The results are then validated using the information of all the model bin sizes within the 0.01 μm to 1 μm range. The fitted distributions on the model outputs are shown in Figure 7 for altitudes between 19 km and 26 km to include the two altitudes (20 and 25 km) that are being investigated because of the irregular altitude grid employed in the GEOS model. Here, the minimized $\chi^2$ of each fit is computed to include all the bins that were omitted during the fitting process. Comparing the magnitudes of the minimized $\chi^2$ values between the two fitted distributions, the gamma distribution provides a better fit to the normalized CARMA model output at all altitudes that were considered in this study. Figure 7 illustrates the difficulties of a UMLN size distribution, which has the tendency to be too wide, and is one of the reasons why generally bimodal distributions have been found to do a better job in representing the OPC

data (Deshler et al., 2003) when there are enough measurements. The gamma distribution does not have the same tendency to overestimate the larger particles. This is confirmed by performing a $\chi^2$ statistic test as to which of these two distribution was a better fit to these model results. The computed percentile values shown in Figure 8 indicate that at each altitude considered, the gamma distribution is the best fit to the CARMA model results, within 15% at the outside. When the minimized $\chi^2$ of each fit is computed applying only the bins that were used for the fitting process, the results (not shown) indicate an decrease in the $\chi^2$ values for both distributions and a resulting increase in the percentile values. The relative differences (RD) computed as percentages using Equation (7) between the phase functions derived from the UMLN $P_u$ and the gamma $P_g$ fits are shown in Figure 9. This provides an indication of the differences that may occur in phase functions from using different size distributions across the range of scattering angles used by LS instruments.

$$RD\,(\%) = \left( \frac{P_g - P_u}{P_u} \right) \times 100\% \tag{7}$$

Finally, a comparison is made between the mean phase functions derived from OPC data fitted with the UMLN distribution and the CARMA model results fitted with the UMLN and gamma distributions at altitude 25 km. The mean and the standard deviation of OPC UMLN phase functions are obtained for each angle from the phase functions of all the months of June, July and August (JJA) from 2008 to 2017. Results from this comparison as shown in Figure 10 indicate that the phase function derived from the gamma distribution fit to the CARMA model outputs at Wyoming agrees very well to, within one standard deviation of, the mean phase function of the JJA UMLN distribution fit to the OPC dataset at this altitude. This agreement between the two phase functions is also shown to be within $\pm 15\%$ at all scattering angles and this reduces to $\pm 5\%$ within the scattering angle range of $15°$ to $180°$. The phase functions derived from the CARMA model outputs using the UMLN distribution are also shown be within one standard deviation of the mean phase function of the JJA UMLN distribution fit to the OPC dataset at this altitude for all scattering angles greater than $15°$. This corresponds to a $\pm 20\%$ change in the phase function for scattering angles greater than $15°$. The good comparison shown by the phase functions derived from the CARMA model with those from the OPC dataset at Laramie, Wyoming, provides evidence for the agreement of the CARMA model with the OPC measurements. This provides a justification for the use of the CARMA model results at other locations on the Earth and for periods with moderate volcanic activity.

The OMPS Version 1.5 (see Table 1) stratospheric aerosol extinction retrieval algorithm uses an ASD which is based on the gamma distribution function that has been derived from the CARMA model outputs at Laramie, Wyoming. The significance of the $P_a(\Theta)$ on LS retrievals as shown by Chen et al. (2018) was done by perturbing each the gamma distribution parameters of the OMPS Version 1.5 ASD and then studying the effect on the retrieved aerosol extinction for the range of scattering angles viewed during a single OMPS/LP orbit. The results showed that a $\pm\,10\%$ change in the gamma distribution parameter $\beta$ would produce a $\pm\,10\%$ change in the calculated $P_a(\Theta)$ at scattering angles between $70°$ and $100°$, whereas a $\pm\,10\%$ change in $\alpha$ results in a $\pm\,3\%$ change in $P_a(\Theta)$ for $\Theta > 70°$. The changes in the retrieved aerosol extinction as shown in Figure 3 of Chen et al. (2018), were found to be approximately anti-correlated with the phase function variation. That is, the fractional change

in the retrieved aerosol extinction was about half of the change in the $P_a(\Theta)$ depending on the single scattering angle. This showed that underestimating the $P_a(\Theta)$ would overestimate the retrieved aerosol extinction and vice-versa.

Additionally, the relative differences of the extinction profiles derived using the gamma distribution function and compared with collocated zonal mean profiles of SAGE III (on the International Space Station) extinction at 675 nm for the months of June to December 2017 have shown to be in agreement within generally less than 10% for altitudes 19 -29 km, with larger differences observed below 18 km due to uncertainties in the LP aerosol retrievals (see figure 12 of Chen et al. (2018)). The improvement observed in the aerosol extinction retrievals between the OMPS V1.0 and V1.5 is a source of motivation for a future OMPS/LP aerosol retrieval algorithm where the CARMA model results would be used to include the variation of the ASD and the $P_a(\Theta)$ with season, latitude, altitude and after a volcanic eruption. The current algorithm assumes that these properties do not vary with altitude and location (Chen et al., 2018).

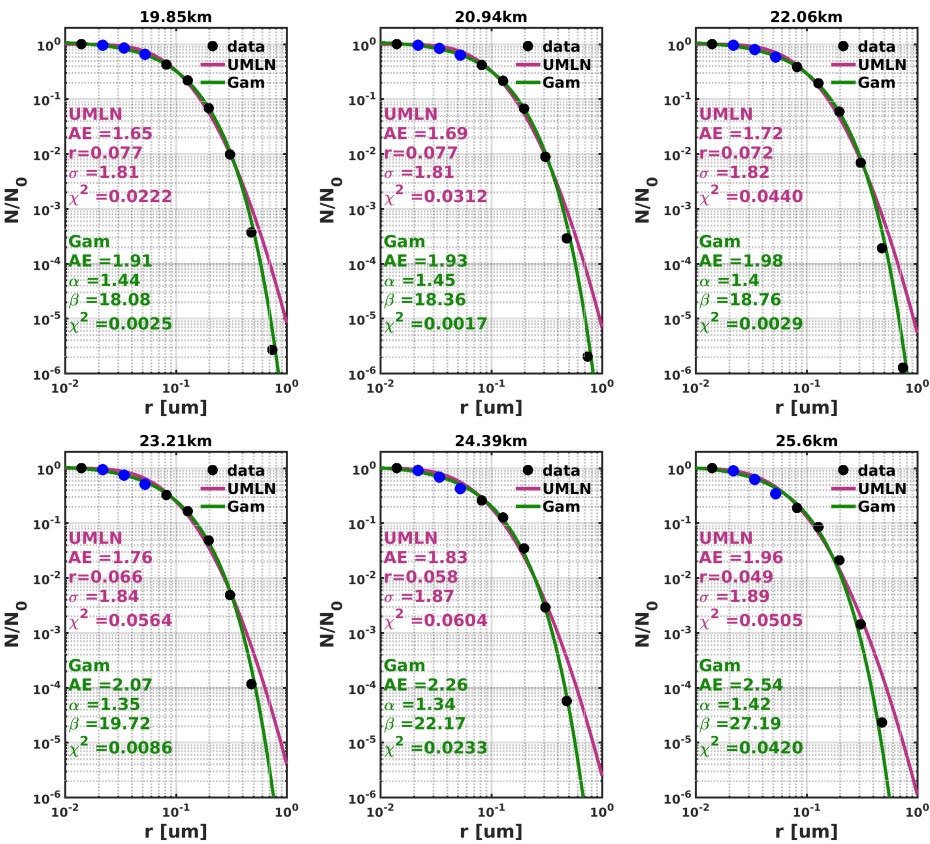

**Figure 7.** Unimodal lognormal and gamma distribution fits to the normalized CARMA model data. The blue data points are excluded during the fitting procedure, but are included during the validation of the fits. The green lines are the gamma distribution fits and the purple lines are the UMLN distribution fits.

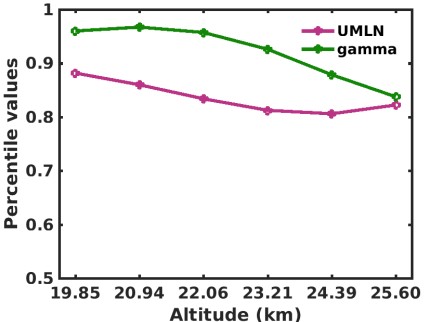

**Figure 8.** Percentile values computed from the minimized $\chi^2$ of the fits of both the UMLN and gamma distributions to determine the level of confidence for which either distribution is chosen to describe the CARMA model data.

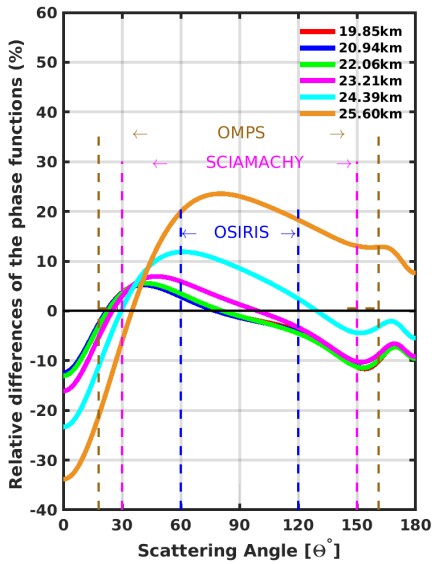

**Figure 9.** Relative differences between the phase functions derived from the gamma and the UMLN parameters fitted to the CARMA model data at each of the six altitudes from 19.85 km to 25.60 km.

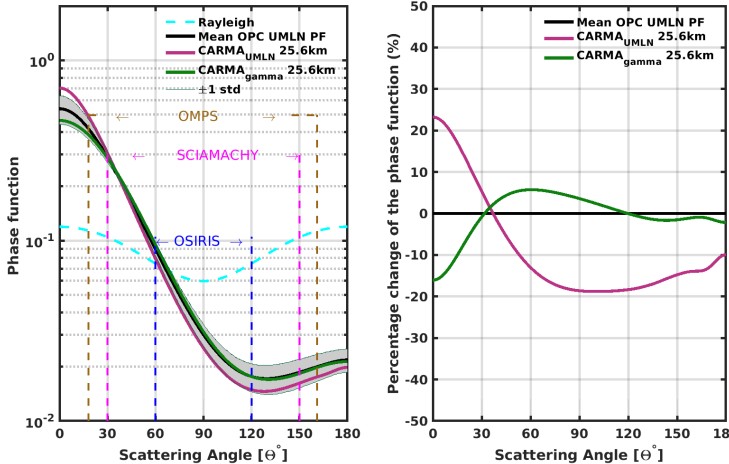

**Figure 10.** The panel on the left shows the mean and the standard deviation of phase functions, for the wavelength of 675 nm, derived from the fits of the UMLN distribution for all the OPC data used for the months of June, July and August (JJA) at altitude 25km compared to the phase functions derived from the UMLN and gamma fits to the CARMA model data at altitude 25.6 km. The panel on the right shows the percent changes between the UMLN derived phase functions from the OPC data and the UMLN and gamma derived phase functions from the CARMA model outputs.

## 4   Concluding discussions and Summary

Measured limb scattered radiance is sensitive to presence of stratospheric aerosols due to the long path the scattered solar photons have to travel through the aerosol layer to reach the sensor (Rieger et al., 2015; Loughman et al., 2018; Chen et al., 2018). This radiance is composed of photons that were singly scattered directly along the line of sight (LOS) of the instrument

and photons that were scattered multiple times before they were finally scattered into the LOS of the instrument. Along the LOS of the sensor, the scattered radiance arises (in part) from the scattering by aerosols, whose angular distribution is described by the phase function, $P_a(\Theta)$, but is attenuated by molecular scattering and trace species absorption, making untangling of the information content in these measurements very complicated. Moreover, diffuse upwelling radiation from the lower atmosphere is also scattered and attenuated along the LOS. To unravel the composition of these measurements requires a good knowledge

of $P_a(\Theta)$ which is derived through the aerosol size distribution (ASD) that is assigned to these aerosols, and thus the choice of which theoretical distribution should be used to describe these particles in the stratosphere is important.

We have investigated fitting a unimodal lognormal (UMLN) and a gamma distribution to the 2008 to 2017 Wyoming in situ LPC measurements, which include a measurement below 0.1 μm, for altitudes 20 km and 25 km. The parameters of the distributions were found by minimizing the $\chi^2$ test statistic between the measurements and the theoretical distributions. As a

first step, we assumed that the stratospheric aerosol is distributed with a single mode during the background conditions and could be fitted to either of the two distributions. Typically, both cumulative distributions are found to be good representatives

for these measurements as was suggested by the $\chi^2$ values. To discriminate between them, a $\chi^2$ goodness of fit test applied showed that to a 10% level of confidence the UMLN was the better of the two distributions as it fitted all data at the two altitudes and for all the months of data that were considered.

Additionally, it has been shown that when the same LPC concentration measurements are fit without using the 0.092 μm bin, the gamma distribution provides a somewhat better fit to particles of radii between 0.01 μm and 0.1 μm range when compared to the UMLN distribution; however the gamma distribution in both cases underestimates the concentrations of the larger particles, corresponding to $\pm22\%$ within the scattering angle ranges of limb sounding satellites for the 675 nm wavelength, when the $P_a(\Theta)$ derived from the both gamma distributions are compared to the $P_a(\Theta)$ derived from the UMLN distribution which estimated these particles exactly. This limited analysis suggests that when a single mode ASD was fitted to aerosol data that did not include sizes below 0.1 μm then the gamma distribution provided the better fit. When particle measurements below 0.1 μm were included, the UMLN distribution provided the better fit to the data.

A similar analysis was further conducted using data obtained from the aerosol microphysical model, CARMA to ascertain which distribution was the best to represent the background aerosol load in the stratosphere. Again, both distributions fitted these data very well for all the altitudes considered. Quantitative comparisons of the goodness of fit for these unimodal distributions indicated that the gamma distribution does a slightly better job for these comparisons in a volcanically quiescent stratosphere. This corresponded to $\pm25\%$ in the computed relative differences in the phase functions at all altitudes between the two distributions. These kinds of closure studies are important for the improvement of confidence levels in space-based data that is used to test aerosol microphysical models and for estimating radiative forcing due to stratospheric aerosols.

The overall implication of this study is to show the importance of the nature of $P_a(\Theta)$ used in the retrieval of the stratospheric aerosol extinction from limb scattering measurements. Typically the phase function is derived from the parameters of a UMLN or a gamma distribution fitted to in situ data which may or may not include a measurement below 0.1 μm radius. The work here shows the differences which can occur between fits made using a UMLN distribution and fits made using a gamma distribution. This leads to some disparity in the phase functions used to represent the measurements. Thus, it is imperative for one to have a knowledge about the nature of the measurements from which the parameters of any distribution are provided.

*Data availability.* OPC data are available for download at University of Wyoming (2018).

*Competing interests.* The authors declare that they have no conflict of interest.

*Acknowledgements.* This work was supported by NASA GSFC through SSAI Subcontract 21205-12-043. The Wyoming in situ data are collected through support from NSF under awards: 0538679, 1011827, 1619632. The authors thank NASA and NOAA for supporting limb

scattering research, and particularly recognize Didier Rault for years of leadership developing the OMPS LP algorithms. We would also like to thank Matthew DeLand for his support.

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
