# Peer review of "A comparison of lognormal and gamma size distributions for characterizing the stratospheric aerosol phase function from OPC measurements"

_Atmospheric Measurement Techniques, 2019_

## Referee Comment (RC1) · Anonymous Referee #1 · 19 Apr 2019

General comments:

This manuscript deals with the nature of the particle size distribution of stratospheric sulfate aerosols. The main motivation is to improve assumptions on the aerosol scattering phase function required to retrieve aerosol extinction coefficients from satellite limb-scatter measurements. The study presents a re-analysis of balloon-borne measurements of the aerosol size distribution with optical particle counters. Specifically, two different size distributions (uni-modal log-normal and gamma distributions) are used to model the observed cumulative distributions. The manuscript is interesting,

presents relevant new information and should eventually be published in my opinion. The paper is very well written and generally easy to follow. There are several points I ask the authors to consider. Specific comments (often minor) are listed below. In addition, I have one more general comment:

The analysis is based on a more limited number of OPC channels than previous analyses of the measurements. In particular, the channels corresponding to large particle sizes are now not considered. These channels provided evidence for a second mode of the particle size distribution, even under background conditions. The second mode is now entirely neglected and the reader wonders, whether the authors now believe that the second mode does not really exist? I think this aspect should be explicitly addressed in the paper. The small number of large particles contributes substantially to the overall aerosol scattering signal and will probably also have a non-negligible effect on the scattering phase function. This is particularly relevant, because the gamma distribution systematically underestimates the number of particles for the largest size bin (top right panel of Fig. 6.).

Specific comments:

Page 2, line 31: "using Mie theory (Deirmendjian, 1969)"

I suggest citing the original paper by Mie here (Mie, 1908).

Page 2, same line: "Here we make the assumption that the aerosol particles in the stratosphere are spherical .."

If Mie theory is used this assumption is implicitly made anyway. Perhaps this could be explicitly stated.

Page 3, line 61: "to correct the ASD" -> "to retrieve the ASD" ?

Page 3, line 66: "and found out that even if the particles were assumed to be spherical .."

I don't understand this part of the sentence, because (a) if Mie theory is used the particles are implicitly assumed to be spherical anyway, (b) if the HG phase function is used no explicit assumptions on the particle shape have to be made, right?

Page 4, lines 108/109: coagulation is certainly also an important process for the growth of stratospheric sulfate aerosols.

Page 6, line 162 and equation (3): If  $O_i$  is already the "frequency" in each size bin, i.e. normalized to the total number of observations, then the multiplication of the expected probability values \zeta\_i with n in equation (3) is not required, is it?  $O_i$  corresponds to a probability then, and so does \zeta i.

Page 7, lin 185: "The LPC data consists of 20 months"

Table 2 lists more than 20 months.

Page 10, Figure 1: I think it would be quite interesting for the reader to see plots of the non-cumulative versions of the gamma and UMLN distributions for these cases.

Same Figure: It is also worth mentioning in the text that two ASDs differ substantially for radii > 300 nm. At 600 nm or so the difference reaches one order of magnitude.

Page 11, line 242: "This is shown in Figure 5, where one observes a considerable change in the magnitude of the phase function, especially in the back-scattering directions ( $\$  theta >= 90) for this X value"

I don't think this statement is correct. Looking at the Figure, the phase function for X=1 is almost constant for scattering angles > 90 deg. Perhaps you intended to make another point?

Page 12, caption Figure 5: "increase .. complexity of the phase function"

The complexity (e.g. for X=10) is mainly due to the fact that a monodisperse aerosol is assumed here. If you assumed a UMLN or a gamma distribution then the oscillations will be damped.

СЗ

Page 13, Figure 6 and related discussion in the main text: I certainly agree that the differences between the OPC-like and LPC-like fits are smaller for the gamma distribution than for the UMLN distribution. However, both gamma distributions systematically underestimate the number in the largest size bin. If larger size bins would be considered this low bias would probably be even larger. So, the two gamma distributions are in good agreement, but they are also both systematically wrong. Perhaps their phase functions deviate even more from the actual phase function compared to the phase function based on the UMLN distributions? Looking at Chi2, the UMLN distribution without the extra measurement still shows the best performance. I'm not asking for any more analysis here, but I think it should be clearly stated that the gamma distributions fail to capture the OPC measurements for the largest sizes, which will lead to a systematic error in the derived phase functions.

Page 14, line 287: "The gamma distribution does not have the same tendency to overestimate the larger particles."

This is now different from the earlier analysis of the OPC/LPC measurements, where the gamma distribution systematically underestimated the large particles.

Page 16, Figure 7: "The blue data points"

I can't identify blue points on my printout.

Page 18, line 325: "Additionally, it has been shown that whenever OPC-like concentration measurements are made, the gamma distribution is the best distribution to be fitted"

I don't agree with this statement, because chi2 for the OPC-like measurements is significantly smaller for the UMLN distribution than for the gamma distribution. Please rephrase this statement to eliminate this apparent contradiction. As mentioned above, the difference between the gamma-fits for the OPC-like and LPC-like measurements is admittedly very small, but the gamma distribution systematically underestimates the

measurements for radii > 300 nm. Since the large particles dominate the scattering signal, they will have a non-negligible effect on the phase function. It may even be possible that the OPC-like UMLN distribution yields a phase function that agrees best with the actual phase function.

Page 18, general comment on the conclusions: the 2nd mode reported in earlier studies is now entirely neglected. The earlier OPC measurements showed indications for the second mode even under background conditions. I guess these measurements are still valid – they are also based on more channels at larger radii. It would be good if the authors would comment on how to treat the second mode in future studies. The large particles with radii of several 100 nm may have a substantial impact on the overall scattering properties and the phase functions of stratospheric sulfate aerosols.

Typos etc.:

Page 2, line 7: "Philippines, 1991" -> "Philippines, 1991"

Page 2, line 35: ",longitude" -> ", longitude"

Page 3, line 58: "occulation" -> "occultation"

Page 3, line 58: "was began" -> "was begun"

Page 3, line 59: I think "that" in "that provided" can be omitted.

Page 3, line 67: "calculations .. was" -> "calculations .. were"

General comment on spelling of "Angström": sometimes you use "A" as the first letter, sometimes "\AA". I think the latter is correct and should perhaps be used throughout the manuscript.

Page 3, line 75, equation (1): "nm" can be omitted here (4 occurrences)

Page 4, line 86: "on measurements from Laramie, Wyoming optical particle counter (OPC) measurements"

Please rephrase

Page 4, line 98: "by (Deepak .." and next line "or (Hinds"

Wrong cite command used (\citep -> \cite}

Page 4, line 10: "Sparc" -> "SPARC"

Page 7, line 183: "following (Kovilakam"

\citep -> \cite

Page 9, line 226: add space after "shape parameter"

Page 12, line 260: ".This" -> ". This"

Page 14, lines 264 and 266: \citep -> \cite

Page 18, line 309: "in the along the" -> "along the"

Page 23, line 471: "Sparc" -> "SPARC"

Same line: add space in "(eds.), SPARC"

---

## Referee Comment (RC2) · Anonymous Referee #2 · 29 May 2019

**General comment:**

In the present paper, authors try to answer the question which shape of the aerosol size distribution (ASD) is it better to use for stratospheric aerosols. In the paper, two shapes of the stratospheric ASD were taken into consideration, namely, uni-modal lognormal (UMLN) and gamma-distribution. Both distributions were fitted to the data from Optical Particle Counters (OPC) and the CARMA model. The quality of the fits was compared using the $\chi^2$ criterion. Based on this comparison, it was concluded that

gamma-distribution provides more realistic aerosol phase function (APF) than UMLN distribution. The latter application is particularly important for the aerosol extinction retrievals from the limb scatter instruments.

While the research itself is thoroughly conducted and convinces the reader that gamma-distribution fits better than UMLN OPC and CARMA model data, the part about the use of gamma-distribution in the limb scatter retrievals is completely missing. There is a long discussion in the manuscript about the importance of the APF for limb scatter instruments (which is absolutely true), and there are nice studies showing the APF from the gamma-distributions. However, the authors did not show any application of the improved APF in the retrievals. Based on this major issue, the following can be suggested:

- authors include some additional study, where the improvement of the limb retrievals with the corrected APF is shown;

- or authors revise the manuscript in a way that they, for example, leave the recommendation to fit OPC data with gamma-distribution rather than with UMLN during the background aerosol loading.

While both revisions will be sufficient to publish the manuscript in AMT, I would suggest going with the first one. Otherwise, the purpose for the APF discussion should be justified differently.

**Specific comments:**

**P.2, L.1:** Maybe it would be good to add to the cited works the newer studies? E.g., Ivy, D. J., Solomon, S., Kinnison, D., Mills, M. J., Schmidt, A., and Neely, R. R.: The influence of the Calbuco eruption on the 2015 Antarctic ozone hole in a fully coupled

chemistry-climate model, Geophysical Research Letters, 44, 2556–2561, 2017.

**P.2, L.27:** Here it is important to mention such sources of stratospheric aerosols as wildfires smoke (see for example Khaykin et al. (2018), https://doi.org/10.1002/2017GL076763) and $SO_2$ from Asian pollution (e.g., Randel et al. (2010), DOI:10.1126/science.1182274).

**P.2, L. 28 and 33:** Is there a difference between $P_a(\Theta)$ and APF? If there is, then it should be better highlighted. If there is not, then just one abbreviation should be used throughout the manuscript.

**P.3, L.70:** It would be nice to mention here, and in Table 1 SCIAMACHY aerosol extinction algorithm V1.4 (see Rieger et al. (2018)).

**P.3, Eq.(1):** The above-mentioned products provide aerosol extinction at one wavelength, so the Eq. (1) can not be used for them to calculate Ångström exponent, because the second extinction coefficient is missing. However, the Eq. (1) is generally absolutely correct and can be used to calculate Ångstöm exponent using the ASD and Mie theory. It would be better to add the sentence before, that the formula is correct for the general case. Otherwise, the reader gets the impression that Ångström exponent is computed from the products.

**P.4, L.103:** Firstly, for all three publications cited here UMLN was used. Secondly, they all used certain assumptions (simply because spaceborne measurements do not provide enough pieces of information). I think it should be mentioned here.

**P.6. L.172:** I think it should be explained why the particles in size range between 0.05 and 0.1 $\mu$m are so important in this study. Smaller particles also scatter solar radiation, and the next sentence says that OPC measurements include particles with radii greater than 0.01 $\mu$m. Therefore, the importance of this particular size range should be justified.

**P.12, L.258-261:** It is hard to understand the purpose of the whole Section 3.2 and its main message. Is the purpose to show that gamma-distribution is less sensitive to the particles smaller than 0.1 $\mu$m? Then it is a good result for OPC fit, and it should be highlighted. However, for the limb instruments, this fit might be relatively useless then.

[Figure]

Coarse resolution of the data on particles smaller then 0.1 $\mu$m does not mean that there are no particles of this size and that they will not influence the "real" distribution. Or is there a misunderstanding of the Section?

**P.14, L.272:** Firstly, it is better to use $\mu$m instead of the nm here, because it might confuse the reader. Secondly, I assume that the bins are not equally distributed over the presented size range and that there is information on small particles. Were there attempts to fit gamma-distribution to the "raw" output of CARMA model to see how this distribution behaves with more information on the particles smaller than 0.1 $\mu$m? Or this question is irrelevant because the purpose of Section 3.2 was wrongly interpreted?

**P.15, L.303-305:** If I understand correctly, CARMA is planned to be used for OMPS retrieval, which should be explicitly mentioned.

**P.18, L.334-349:** As it was said in the general comments, the part about the space-borne instruments is absolutely missing. Thus, it should be either removed and reformulated for OPC measurements, or some real studies using limb instruments should be done.

**Technical corrections:**

**P.1, L.1-2:** The first sentence in the abstract leaves an impression that OPC provided measurements only from 2008-2017, which is not true. See e.g., Deshler et al. 2003.

**P.3, L.28:** There is not much sense to shorten "solar occultation" to "SO" since it is used just once. If the authors want to save some space, it is better to shorten "Figure" to "Fig." and "Equation" to "Eq.".

**P.4, L.98-99:** The citation here should be done as "Deepak and Box (1982) or Hinds (1982)".

**P.4, L.101:** Sparc better spelled as SPARC.

**P.6, L.151:** Here I think is a typo, and 6 data points were meant.

**P.8, L.212:** Maybe "percentile" should not be in italics?

**P.14, L.282:** Maybe leave $\chi^2$ here instead "chi-squares"?
**P.18, L.308:** I think citations should be listed chronologically.

---

## Referee Comment (RC3) · Anonymous Referee #3 · 4 Jun 2019

This paper presents an analysis of the suitability of log-normal and gamma distributions to the particle size measurements from in situ OPC balloon flights. The authors motivate this work based on the implications that the fitted distribution has on the derived aerosol scattering phase function that is required in the radiative transfer forward modelling for limb scattering retrievals of aerosol extinction.

The results have merit and the study is well conducted; however, I completely agree with the major issue raised by Referee #2. The study needs to include a quantitative assessment of the impact these results have on the aerosol retrievals. Reporting the

difference in phase functions, as the study currently stands, is of limited use, but with some additional work to show the impact on the retrievals, it becomes potentially quite important. One aspect to consider for example is that the forward scattering peak that the authors sometimes choose to cut off the phase function figures can be quite important with multiple scattering and high albedo. In line with this comment, I think the authors should put this study more deeply in the context of the Chen et al., 2018. There are similarities and those should be discussed in detail in light of the new results. Finally, the work would be more broadly useful if wavelengths other than 675 nm were also studied (SCIAMACHY and OSIRIS use 750 nm for example).

Minor comments:

Mixed use of APF and P_a in the text for the aerosol phase function. Choose one.

Abstract line 11: what does "stable" mean?

Abstract last sentence: The exclusion of certain bins is too specific for the nature of the rest of the abstract (cannot be understood without a lot more detail from the paper)

Introductory paragraph should probably contain some motivating statement about the impact of several moderate volcanic eruptions over the last ∼decade.

Line 32: what does "homogeneous" mean? i.e. there is still a size distribution of particle sizes; also, the refractive index should be for hydrated sulfuric acid, and should be stated and referenced.

Lines 65-68: Quantify "sufficient" and "high precision"; this statement needs more detail

Line 69: Bourassa et al., ACP, 2012 is the reference for OSIRIS version 5.0

Line 72: Size distribution parameters for OMPS v1.0 and v1.5 should be stated, possibly included in Table 1 somehow

Line 73: Use of Angstrom exponent should be motivated; this statement is out of place at the moment

Equation 1: Typesetting with units is strange

Line 159: "similarity in appearance" needs quantification; otherwise this is not a helpful statement

Line 163: No brackets on equation numbers

Table 2: Is this information necessary?

Figure 1: Green text on figures is hard to read

Line 218: something wrong with the wording here

Line 223: It doesn't follow that the phase functions agree for scattering angles greater than 20 degrees "because the fits of the two distributions overlap"

---

## Author Comment (AC1) · 26 Sep 2019

**General comments:**

This manuscript deals with the nature of the particle size distribution of stratospheric sulfate aerosols. The main motivation is to improve assumptions on the aerosol scattering phase function required to retrieve aerosol extinction coefficients from satellite limb-scatter measurements. The study presents a re-analysis of balloon-borne measurements of the aerosol size distribution with optical particle counters. Specifically, two different size distributions (uni-modal log-normal and gamma distributions) are used to

model the observed cumulative distributions. The manuscript is interesting, presents relevant new information and should eventually be published in my opinion. The paper is very well written and generally easy to follow. There are several points I ask the authors to consider. Specific comments (often minor) are listed below. In addition, I have one more general comment:

The analysis is based on a more limited number of OPC channels than previous analyses of the measurements. In particular, the channels corresponding to large particle sizes are now not considered. These channels provided evidence for a second mode of the particle size distribution, even under background conditions. The second mode is now entirely neglected and the reader wonders, whether the authors now believe that the second mode does not really exist? I think this aspect should be explicitly addressed in the paper. The small number of large particles contributes substantially to the overall aerosol scattering signal and will probably also have a non-negligible effect on the scattering phase function. This is particularly relevant, because the gamma distribution systematically underestimates the number of particles for the largest size bin (top right panel of Fig. 6.)

**Response:** The fitting of two uni-modal distributions to OPC and OPC$_{nsb}$ data was motivated by Figure A2 of (Chen et. al 2018), who fitted four Bi-modal lognormal distributions (BMLN) to OPC$_{nsb}$ data measured on 12 April 2000 for altitude 20 km. All four fits had a similar AE of approximately 2.4, but each had different coarse mode fraction (CMF). These four BMLN distribution fits to the OPC data differed significantly from each other in the radius range between 0.01 µm to 0.1 µm and these differences resulted from the gaps in the OPC size bins that limited the ability of the fits to be constrained. All four fits captured the larger particles very well but the resulting phase functions differed from each other because of the overestimation or under estimation of the particles between 0.01 and 0.1 µm. The two gamma distributions are in good agreement with each other especially within the particle radius range of 0.01 µm to 0.1 µm but both systematically failed to capture the number concentration of the largest

bin. Since the actual phase function is not known, it is possible that either of the phase functions from the UMLN or gamma may be right.

Also it should be noted that most of the OPC measurements associated with the OMPS measurement period lack measurable signals in the larger bins, especially in the 20-25 km altitude range that is most relevant for the current OMPS retrieval assessments (Chen et. al 2018). Therefore, these measurements do not provide a clear argument for the presence of larger particles in those cases. For cases that DO have a measurable signal for large aerosol bins, the signal remains much lower than in the smaller size bins. So (in a global, weighted fit like was performed), it may be appropriate for the fit to the data at the largest size bins to be very poor (relative to the the much higher signals in the smaller bins).

**Specific comments:**

**1-** Page2, line 31 "using Mie theory (Deirmendjian,1969)" I suggest citing the original paper by Mie here (Mie,1908)

**Response:** The line has been updated to "using Mie theory (Mie,1908)"

**2-** Page 2, same line "Here we make the assumption that the aerosol particles in the stratosphere are spherical" If Mie theory is used this assumption is implicitly made anyway. Perhaps this could be explicitly stated.

**Response:** This sentence has been revised to: "Theoretically, the $P_a(\Theta)$ is calculated from the aerosol size distribution (ASD) using Mie theory (Mie, 1908), generally assuming that the aerosol particles in the stratosphere are spherical and homogeneous."

**3-** Page3, line 61: "to correct the ASD"
**Response:** This sentence has been revised to read: "to retrieve the ASD"

**4-** Page3, line 66: "and found out that even if the particles were assumed to be spherical" I don't understand this part of the sentence, because, (a) if Mie theory is used the

particles are implicitly assumed to be spherical anyway, (b) if the HG phase function is used no explicit assumptions on the particle shape have to be made, right?

**Response:** The statement has been revised, it now reads: "Some techniques that have been used to model the $P_a(\Theta)$ are by computing it using the Henyey-Greenstein phase function (H-G) (Henyey and Greenstein, 1941; Ernst, 2013; Grams, 1981) or the modified Henyey-Greenstein phase function (MH-G) (Irvine, 1965; Cornette and Shanks, 1992) with a precise asymmetry factor g, which is the average cosine of the scattering angle weighted by the phase function. The shortcomings of using these functions to approximate the real Mie phase function were demonstrated by Toublanc (1996) for two cases. When the radius of the particle was ten times smaller than the wavelength, the H-G phase function failed to produce the shape of the real Mie phase function in comparison to that of the MH-G. By contrast, for a particle of radius that was comparable to the wavelength, both functions failed to reproduce the lobe patterns of the real Mie phase function."

**5-** Page4, lines108/109: coagulation is certainly also an important process for the growth of stratospheric aerosols.

**Response:** Coagulation is also an important stratospheric aerosol formation process and it has been included in the sentence, which now reads:

'A multimodal distribution can be used to represent coexisting "nucleation" , "coagulation", and "accumulation" modes after a volcanic eruption. The nucleation mode is associated with new particle formation from sulfur vapor which quickly coagulate to form larger particles (Hamill et al., 1997), and the accumulation mode associated with particle growth by condensation of the vapor on the existing particles (Steele and Turco, 1997).'

**6-** Page6, line 162 and equation(3): if $O_i$ is already the "frequency" in each size bin, i.e. normalized to the total number of observations, then the multiplication of the expected probability values $\zeta_i$ with n in equation (3) is not required, is it? $O_i$ corresponds to a

probability then, and so does $\zeta_i$

**Response:** $O_i$ is the "frequency" in each size bin. It is normalized by dividing by "n". Rearranging the equation will then lead to the "n" in the denominator being squared ( which was omitted). Equation (3) has now been updated squaring "n" in the denominator. On the other hand, if $O_i$ is defined as the normalized "frequency" in each size bin, then the multiplication of the expected probability values $\zeta_i$ with "n" in the equation would not be required.

**7-** Page7, line 185: "The LPC data consists of 20 months of measurements" Table 2 lists more than 20 months.

**Response:** This sentence has been corrected to read: "The LPC data consists of 27 months of measurements" to correspond to the number of months listed in Table 2.

**8-** Page,10 Figure 1: I think it would be quite interesting for the reader to see plots of the non-cumulative versions of the gamma and UMLN distributions for these cases.

**Response:** Figure 1 has been updated to include non-cumulative versions of the gamma and UMLN distributions for the two cases shown.

**9-** Same Figure: It is also worth mentioning in the text that the ASDs differ substantially for radii > 300 nm. At 600 nm or so the difference one order of magnitude.

**Response:** The text has been updated to include the above suggestion:

"The two ASDs tend to diverge beginning at radii greater than 300 nm and differ substantially at approximately 600 nm, where aerosol concentrations are below the minimum detectable concentrations, and there these differences can reach one order of magnitude."

**10-** Page 11, line 242: "This is shown in Figure 5, where one observes a considerable change in the magnitude of the phase function, especially in the back-scattering directions ($\Theta \geq 90°$) for this X value" I don't think this statement is correct. Looking

at the Figure, the phase function for X=1 is almost constant for scattering angles > 90 degrees. Perhaps you intended to make another point?

**Response:** The above statement has been rephrased to read:

"The phase function for X = 3 shows a forward peak and is nearly constant for scattering angles ($\Theta \geq 70\ °$). When there are no measurements between the 0.01 and 0.15 µm bin sizes, then the particle concentration within this range is estimated by the function used to fit the data. Errors in estimating the number of particles within this range by the function used for fitting the data will lead to uncertainties in the phase function as shown by the X = 1 plot in Figure 5.

**11-** Page 12, caption Figure 5: "increase .. complexity of the phase function" The complexity (e.g. for X=10) is mainly due to the fact that a mono disperse aerosol is assumed here. If you assumed a UMLN or a gamma distribution then the oscillations will be damped.

**Response:** The caption of Figure 5 has been revised to read: "Mie phase functions of a monodisperse aerosol for different values of the size parameter X derived with a refractive index of 1.33. The increasing asymmetry and complexity (e.g. for X=10) of the phase functions with increasing X is due to the use of a monodisperse aerosol. The oscillations observed are damped when the phase functions are computed for an ensemble of aerosols that are assumed to have a UMLN or gamma distribution. The phase functions are shown for the range of scattering angles that are observed by OMPS, SCIAMACHY and OSIRIS."

**12-** Page 13, Figure 6 and related discussion in the main test: I certainly agree that the differences between the OPC-like and LPC-like fits are smaller for the gamma distribution than the UMLN distribution. However, both gamma distributions systematically underestimate the number of in the largest bin. If larger bins would be considered this low bias would probably be even larger. So the two gamma distributions are in good agreement, but they are also both systematically wrong. Perhaps their phase functions

deviate even more from the actual phase functions compared to the phase function based on the UMLN distribution? Looking at the $\chi^2$, the UMLN distribution without the extra measurement still show the best performance. I am not asking for any more analysis here, but I think it should be clearly stated that the gamma distributions fail to capture the OPC measurements for the largest sizes, which will lead a systematic error in the derived phase functions.

**Response:** "OPC-like" has been changed to OPC$_{nsb}$ and "LPC-like" has been changed to OPC: "nsb" stands for no small bin .

There was no arbitrary decision to ignore measurements from the largest bin when the fits were made. Thus the systematic underestimation of the concentration of the largest bin by the gamma distribution fits was not deliberate as the same can be seen for the UMLN fit (red line). Because the actual phase function is not known, it is possible that the phase functions derived from the gamma or the UMLN distribution functions may be the right one.

This has been stated in the text as "The failure of both gamma distributions to capture the OPC measurements for the largest bin size for the case shown in Figure 6 could lead to a systematic error in the derived phase functions."

**13-** Page 14, line 287: "The gamma distribution does not have the same tendency to over estimate the larger particles" This is now different from the earlier analysis of the OPC/LPC measurements, where the gamma distribution systematically underestimated the large particles.

**Response:** These are the CARMA microphysical model outputs at Wyoming and these are different from the OPC/LPC measurements made at the same location.

**14-** Page 16, Figure 7: "The blue data points" I can't identify blue points on my printout.
**Response:** The size of the blue dots has been increased.

**15-** Page 18, line 325: "Additionally, it has been shown that whenever OPC-like concentration measurements are made, the gamma distribution is the best distribution to be fitted" I don't agree with this statement, because $\chi^2$ for the OPC-like measurements is significantly smaller for the UMLN distribution than the gamma distribution. Please rephrase this statement to eliminate this apparent contradiction. As mentioned above, the difference between the gamma-fits for the OPC-like and LPC-like measurements are admittedly very small, but the gamma distribution systematically under estimates the measurements for radii > 300 nm. Since the large particles dominate the scattering signal, they will have a non-negligible effect on the phase function. It may even be possible that the OPC-like UMLN distribution yield a phase function that agrees best with the actual phase function.

**Response:** Figure 6 is a comparison of the fits made with the inclusion of a small bin (OPC) to one with no small bin (nsb) OPC$_{nsb}$ measurements. On this figure, the $\chi^2$ value for OPC UMLN distribution fit is 0.0135 and that of gamma distribution fit is 0.0101. This shows that the $\chi^2$ for the gamma fit is somewhat less than that of the UMLN fit. This is in agreement with the statement:

"Additionally, it has been shown that when the same LPC concentration measurements are fit without using the 0.092 μm bin, the gamma distribution provides a some what better fit because of its insensitivity to particles between 0.01 μm and 0.1 μm range when compared to the UMLN distribution; however the gamma distribution in both cases underestimates the concentrations of the larger particles, which may be quite important depending on the wavelength of interest"

**16-** Page 18, general comment on the conclusion: the 2nd mode reported in earlier study is now entirely neglected. The earlier OPC measurements showed indications for the second mode even under background conditions. I guess this measurements are still valid- they are also based on more channels at larger radii. It would be good if the authors would comment on how to treat the second mode in future studies. The large particles with radii of several 100 nm may have a substantial impact on overall scattering properties and the phase function of stratospheric sulfate aerosols.

**Response:** During background conditions, Deshler et al. (2003) have in some cases used a bimodal lognormal (BMLN) particle size distribution to achieve the best fit to the OPC measurements made by the in situ optical counters. However, with limb scattering geometry this BMLN size distribution is not possible because six (or five when the data points are normalized) independent pieces of information at each altitude will be needed to describe the BMLN distribution, but at altitudes greater than 20 km, OPC measurements mostly provide four data points. This makes it impossible to fit a bimodal distribution.

Also, we agree that particles with radii larger than 100 nm may have a substantial impact on the overall scattering properties and the phase functions of the stratospheric sulfate aerosol during volcanically active or periods with pyro CB activity. But most of the limb radiance measurements made by OMPS in the last seven years is devoid of any large volcanic activity sufficient enough to inject aerosols into the stratosphere and are mostly considered as background condition. In the future, we hope to compare the phase functions derived from multi-modal aerosol size distributions.

**Typos etc,:**
**1-** Page 2, line 7: "Philippines,1991" -> "Philippines, 1991"
**Response:** This has been corrected.

**2-** Page 2, line 35: ",longitude" -> ", longitude"
**Response:** This has been corrected.

**3-** Page 3, line 58: "occulation" -> "occultation"
**Response:** This has been corrected.

**4-** Page 3, line 58: "was began" -> "was begun"
**Response:** This has been corrected.

**5-** Page 3, line 59: I think "that" in "that provided" can be omitted.
**Response:** This has been corrected.

**6-** Page 3, line 67: "calculations .. was" -> "calculations .. were"
**Response:** This has been corrected.

**7-** General comment on spelling of "Angström": sometimes you use "A" as the first letter,sometimes "Å". I think the latter is correct and should perhaps be used throughout the manuscript.
**Response:** This has been updated everywhere in the manuscript.

**8-** Page 3, line 75, equation (1): "nm" can be omitted here (4 occurrences)
**Response:** This has been corrected for all occurrences.

**9-** Page 4, line 86: "on measurements from Laramie, Wyoming optical particle counter (OPC) measurements"
**Response:** This sentence has been corrected to read "on data from Laramie, Wyoming optical particle counter (OPC) measurements"

**10-** Page 4, line 98: "by (Deepak .." and next line "or (Hinds" Wrong cite command used (\citep -> \cite}
**Response:** This has been corrected.

**11-** Page 4, line 10: "Sparc" -> "SPARC"
**Response:** This has been corrected.

**12-** Page 7, line 183: "following (Kovilakam" \citep -> \cite
**Response:** This has been corrected.

**13-** Page 9, line 226: add space after "shape parameter"
**Response:** This has been corrected.

**14-** Page 12, line 260: ".This" -> ". This"
**Response:** The correction has been made.

**15-** Page 14, lines 264 and 266: \citep -> \cite
**Response:** This has been corrected.

**16-** Page 18, line 309: "in the along the" -> "along the"
**Response:** The correction has been made.

**17-** Page 23, line 471: "Sparc" -> "SPARC"
**Response:** The correction has been made

**18-** Same line: add space in "(eds.),SPARC"
**Response:** The correction has been made.

---

## Author Comment (AC2) · 26 Sep 2019

In the present paper, authors try to answer the question which shape of the aerosol size distribution (ASD) is it better to use for stratospheric aerosols. In the paper, two shapes of the stratospheric ASD were taken into consideration, namely, uni-modal lognormal (UMLN) and gamma-distribution. Both distributions were fitted to the data from Optical Particle Counters (OPC) and the CARMA model. The quality of the fits was compared using the $\chi^2$ criterion. Based on this comparison, it was concluded that gamma-distribution provides more realistic aerosol phase function (APF) than UMLN

distribution. The latter application is particularly important for the aerosol extinction retrievals from the limb scatter instruments. While the research itself is thoroughly conducted and convinces the reader that gamma-distribution fits better than UMLN OPC and CARMA model data, the part about the use of gamma-distribution in the limb scatter retrievals is completely missing. There is a long discussion in the manuscript about the importance of the APF for limb scatter instruments (which is absolutely true), and there are nice studies showing the APF from the gamma-distributions. However, the authors did not show any application of the improved APF in the retrievals. Based on this major issue, the following can be suggested:

- authors include some additional study, where the improvement of the limb retrievals with the corrected APF is shown;

- or authors revise the manuscript in a way that they, for example, leave the recommendation to fit OPC data with gamma-distribution rather than with UMLN during the background aerosol loading.

While both revisions will be sufficient to publish the manuscript in AMT, I would suggest going with the first one. Otherwise, the purpose for the APF discussion should be justified differently.

**Response:** A parallel study by (Chen et al. 2018) have compared retrieved aerosol extinctions using the OMPS/LP V1.0 (bimodal lognormal), V1.5 (gamma distribution) derived from the CARMA model output to the extinction profile derived from SAGE III (on the International Space Station). The results show an improvement in the V1.5 extinctions to within 10% at altitudes 19-29 km. The authors of the paper are including this information and referencing the above paper.

**Specific Comments:**

**1-** P.2, L.1: Maybe it would be good to add to the cited works the newer studies? E.g., Ivy, D. J., Solomon, S., Kinnison, D., Mills, M. J., Schmidt, A., and Neely, R. R.: The influence of the Calbuco eruption on the 2015 Antarctic ozone hole in a fully coupled chemistry-climate model, Geophysical Research Letters, 44, 2556–2561, 2017

**Response:** The citations have been updated to include (ivy et al. 2017). Also the effect on the ozone hole enhancement by the presence of volcanic aerosols associated with Calbuco has been mentioned in the same paragraph.

**2-** P.2, L.27: Here it is important to mention such sources of stratospheric aerosols as wildfires smoke (see for example Khaykin et al. (2018), https://doi.org/10.1002/2017GL076763) and $SO_2$ from Asian pollution (e.g., Randel et al. (2010), DOI:10.1126/science.1182274).

**Response:** Other sources of stratospheric aerosols from wildfire smoke and $SO_2$ from Asian pollution have been mentioned and the Khaykin et al. (2018) and Randel et al. (2010) have been cited.

**3-** P.2, L. 28 and 33: Is there a difference between $P_a(\Theta))$ and APF? If there is, then it should be better highlighted. If there is not, then just one abbreviation should be used throughout the manuscript.

**Response:** There is no difference between $P_a(\Theta)$ and APF. Only $P_a(\Theta)$ will be used to subsequently represent the Stratospheric Aerosol Phase function.

**4-** P.3, L.70: It would be nice to mention here, and in Table 1 SCIAMACHY aerosol extinction algorithm V1.4 (see Rieger et al. (2018)).

**Response:** SCIAMACHY has been mentioned in both places.

**5-** P.3, Eq.(1): The above-mentioned products provide aerosol extinction at one wavelength, so the Eq. (1) can not be used for them to calculate Ångström exponent, because the second extinction coefficient is missing. However, the Eq. (1) is generally absolutely correct and can be used to calculate Ångstöm exponent using the ASD and

Mie theory. It would be better to add the sentence before, that the formula is correct for the general case. Otherwise, the reader gets the impression that Ångström exponent is computed from the products

**Response:** The sentence has been revised to include that the extinction of the two wavelengths are derived using the ASD and Mie theory. "The figures also display for each fit the Ångström exponent (AE) that was computed using Equation (1), where $\lambda_1$ and $\lambda_2$ are 525 nm and 1020 nm respectively.

**6-** P.4, L.103: Firstly, for all three publications cited here UMLN was used. Secondly, they all used certain assumptions (simply because spaceborne measurements do not provide enough pieces of information). I think it should be mentioned here

**Response:** The cited publications have been updated to include Loughman et al.(2018), which used BMLN aerosol size information for the extinction retrievals. Also it has been mentioned that space-borne measurement do no provide enough pieces of information.

**7-** P.6. L.172: I think it should be explained why the particles in size range between 0.05 and 0.1 μm are so important in this study. Smaller particles also scatter solar radiation, and the next sentence says that OPC measurements include particles with radii greater than 0.01 $\mu$m. Therefore, the importance of this particular size range should be justified.

**Response:** In a case study (see Figure A1 of Chen et al., 2018), four bimodal lognormal size distributions were fitted to the same data set which did not have a measurement between 0.05 $\mu$m and 0.1 $\mu$m. The differences observed in the resulting phase functions were due to the differences of the fits at that radius range because all four fits captured the larger bins very well. This shows the importance of aerosol particles within the radius range 0.05 $\mu$m and 0.1 $\mu$m.

**8-** P.12, L.258-261: It is hard to understand the purpose of the whole Section 3.2 and

its main message. Is the purpose to show that gamma-distribution is less sensitive to the particles smaller than 0.1 $\mu$m? Then it is a good result for OPC fit, and it should be highlighted. However, for the limb instruments, this fit might be relatively useless then Coarse resolution of the data on particles smaller then 0.1 $\mu$m does not mean that there are no particles of this size and that they will not influence the "real" distribution. Or is there a misunderstanding of the Section?

**Response:** This section tests the sensitivity of the two unimodal distributions to determine which distribution would accurately predict the amount of particles within the particles radius range of 0.01$\mu$m and 0.1$\mu$m during the period when there are no measurements within this particle radius range (no small bin (nsb) or $OPC_{nsb}$) and when there is at least a measurement within that range (OPC).

Also the conclusion has been rephrased to read:
"The conclusion drawn from this comparison is that the phase functions calculated with the gamma distributions with and without the small bin are comparable to each other to within 10% as compered to those of the UMLN distribution. This signifies that the gamma distribution is relatively insensitive to the addition of an intermediary bin between 0.05 $\mu$m and 0.1 $\mu$m, whereas the UMLN distribution is quite sensitive to this additional information."

**9-** P.14, L.272: Firstly, it is better to use $\mu$m instead of the nm here, because it might confuse the reader. Secondly, I assume that the bins are not equally distributed over the presented size range and that there is information on small particles. Were there attempts to fit gamma-distribution to the "raw" output of CARMA model to see how this distribution behaves with more information on the particles smaller than 0.1 $\mu$m? Or this question is irrelevant because the purpose of Section 3.2 was wrongly interpreted?

**Response:** First, nm has been converted to $\mu$m. Secondly, The CARMA model "raw" outputs were used because they provide enough information on smaller particles

smaller than 0.1 $\mu$m, and for this study these model outputs were subsetted into the OPC measurement bins to find out which of the two uni modal distributions was the best fit to this model output. The conclusion drawn from section 3.2 is to use the gamma distribution to fit $OPC_{nsb}$ data. This section also shows that the gamma distribution is the best fit to the CARMA model outputs. A table showing the distribution of the aerosol size bins used in the CARMA model has been added.

In our next study we plan to fit the gamma distribution to all the "raw" outputs of the CARMA model.

**10-** P.15, L.303-305: If I understand correctly, CARMA is planned to be used for OMPS retrieval,which should be explicitly mentioned.

**Response:** The plan to use phase functions derived from the CARMA model outputs in OMPS retrievals has been stated in the manuscript.

**11-** P.18, L.334-349: As it was said in the general comments, the part about the space borne instruments is absolutely missing. Thus, it should be either removed and reformulated for OPC measurements, or some real studies using limb instruments should be done

**Response:** A parallel study by (Chen et al. 2018) have compared retrieved aerosol extinctions using the OMPS/LP V1.0 (bimodal lognormal), V1.5 (gamma distribution) derived from the CARMA model output to the extinction profile derived from SAGE III (on the International Space Station). The results show an improvement in the V1.5 extinctions to within 10% at altitudes 19-29 km. The authors of the paper are including this information and referencing the above paper.

**Technical corrections:**

**1-** P.1, L.1-2: The first sentence in the abstract leaves an impression that OPC provided

measurements only from 2008-2017, which is not true. See e.g., Deshler et al. 2003.

**Response:** it has been clarified in the abstract that this is a subset of the total data since measurements have been taking place since 1971 (Deshler et al. 2003).

**2-** P.3, L.28: There is not much sense to shorten "solar occultation" to "SO" since it is used just once. If the authors want to save some space, it is better to shorten "Figure"to "Fig." and "Equation" to "Eq.".

**Response:** Noted

**3-** P.4, L.98-99: The citation here should be done as "Deepak and Box (1982) or Hinds(1982)".
**Response:** Noted

**4-** P.4, L.101: Sparc better spelled as SPARC.
**Response:** Noted

**5-** P.6, L.151: Here I think is a typo, and 6 data points were meant.
**Response:** Because we are using the coarse mode fraction (CMF), which is the ratio of the coarse mode concentration to the total, the number of parameters reduces from 6 to 5.

**6-** P.8, L.212: Maybe "percentile" should not be in italics?
**Response:** Noted.

**7-** P.14, L.282: Maybe leave $\chi^2$ here instead "chi-squares"?
**Response:** Noted

**8-** P.18, L.308: I think citations should be listed chronologically.
**Response:** Noted

---

## Author Comment (AC3) · 26 Sep 2019

This paper presents an analysis of the suitability of log-normal and gamma distributions to the particle size measurements from in situ OPC balloon flights. The authors motivate this work based on the implications that the fitted distribution has on the derived aerosol scattering phase function that is required in the radiative transfer forward modeling for limb scattering retrievals of aerosol extinction.

The results have merit and the study is well conducted; however, I completely agree

with the major issue raised by Referee 2. The study needs to include a quantitative assessment of the impact these results have on the aerosol retrievals. Reporting the difference in phase functions, as the study currently stands, is of limited use, but with some additional work to show the impact on the retrievals, it becomes potentially quite important. One aspect to consider for example is that the forward scattering peak that the authors sometimes choose to cut off the phase function figures can be quite important with multiple scattering and high albedo. In line with this comment, I think the authors should put this study more deeply in the context of the Chen et al., 2018. There are similarities and those should be discussed in detail in light of the new results. Finally, the work would be more broadly useful if wavelengths other than 675nm were also studied (SCIAMACHY and OSIRIS use 750nm for example)

**Response:** Chen et al. 2018, have conducted a parallel study where they compared the retrieved aerosol extinction profiles from the OMPS/LP using the V1.0 (bimodal lognormal distribution) and V1.5 (gamma distribution) retrieval algorithms to the extinction profiles derived from SAGE III (on the International Space Station). The results obtained, indicated an improvement in the V1.5 extinction profiles to within 10% at altitudes 19-29 km. The authors of the paper are including this information and referencing the above paper.

In our next study, we plan to include other wavelengths greater than 675nm.

**Minor Comments:**

**1-** Mixed use of APF and $P_a$ in the text for the aerosol phase function. Choose one.
**Response:** There is no difference between $P_a(\Theta)$ and APF. Only $P_a(\Theta)$ will be used to represent the Stratospheric Aerosol Phase function.

**2-** Abstract line 11: what does "stable" mean?
**Response:** The sentence contain the word "stable" has been removed.

**3-** Abstract last sentence: The exclusion of certain bins is too specific for the nature of the rest of the abstract (cannot be understood without a lot more detail from the paper)
**Response:** Noted. The last part of the Abstract has been rephrased.

**4-** Introductory paragraph should probably contain some motivating statement about the impact of several moderate volcanic eruptions over the last decade.
**Response:** This has been noted and we have added a statement about the impact of moderate volcanic eruptions.

**5-** Line 32: what does "homogeneous" mean? i.e. there is still a size distribution of particle sizes; also, the refractive index should be for hydrated sulfuric acid, and should be stated and referenced
**Response:** The word "homogeneous" is used in this line to mean "the particles have the same properties throughout".The refractive index has been stated for hydrated sulfuric acid and referenced.

**6-** Lines 65-68: Quantify "sufficient" and "high precision"; this statement needs more detail
**Response:** More details have been included to elaborate on the statements made by Toublanc (1996).

**7-** Line 69: Bourassa et al., ACP, 2012 is the reference for OSIRIS version 5.0
**Response:** Noted

**8-** Line 72: Size distribution parameters for OMPS v1.0 and v1.5 should be stated, possibly included in Table 1 somehow
**Response:** The size distribution parameters of OMPS v1.0 and v1.5 have been included in Table 1.

**9-** Line 73: Use of Angstrom exponent should be motivated; this statement is out of place at the moment
**Response:** A motivational statement has been included.

[Figure]

**10-** Equation 1: Typesetting with units is strange
**Response:** The units have been removed from the equation.

**11-** Line 159: "similarity in appearance" needs quantification; otherwise this is not a helpful statement
**Response:** The statement "similarity in appearance" has been deleted from the text.

**12-** Line 163: No brackets on equation numbers
**Response:** Noted and corrected.

**13-** Table 2: Is this information necessary?
**Response:** This information is necessary to show the reader the months in which measurement were made each year and also the frequency of measurements throughout the period considered (2008 - 2017).

**14-** Figure 1: Green text on figures is hard to read
**Response:** A darker shade of green has been used on this figure.

**15-** Line 218: something wrong with the wording here
**Response:** The word "taking" has been replaced with "taken".

**16-** Line 223: It doesn't follow that the phase functions agree for scattering angles greater than 20 degrees "because the fits of the two distributions overlap"
**Response:** The statement "because the fits of the two distributions overlap" has been removed.

---

## Referee Report (RR1)

**General comment:**

In this paper, the authors study which aerosol particle size (ASD) distribution fits better OPC measurements and/or CARMA model output. Here, two ASDs were taken into consideration, namely, unimodal lognormal (UMLN) and gamma distribution. The authors also look at how the aerosol phase function ($P_a(\Theta)$) changes with different ASDs and particle size range taken into consideration.

While, in general, this revision of the manuscript got significant improvements in comparison to the previous version, the authors still did not clarify all the points from the previous review. Because of that, I think, another revision of the manuscript should be done. Even though the conclusions were reformulated, they are not yet sufficient to be published. If I understood this paper correctly, its take-home message is "you should know where your ASD came from". This message does not need 21 pages of studies.

Reading this revision of the manuscript, I got an impression that the authors tried to say that UMLN is a good fit for OPC measurements, while CARMA model output is better fitted with gamma distribution. However, both of the distributions are correct depending on the source of information one uses, and there is no truth, which of them should be used in limb retrievals. This is a very important result for the limb community, which looks for the last 15 years for the solution of the aerosol extinction coefficient dependency on $P_a(\Theta)$ problem, and in particular, tries different ASDs in order to improve products. If I am correct in my understanding, then the conclusions should be reformulated this way, providing numerical proof. The authors quantify the changes in the $P_a(\Theta)$; however, they do not explain how this change in $P_a(\Theta)$ influences the resulting limb extinctions, which in my opinion is absolutely essential. Again, if the authors do not want to deal with limb retrievals, that is fine, but then the paper should be revised to remove a long introduction about limb instruments. If the authors want to leave the discussion on how this study is important for the limb community, then they should show it differently. The suggestions on how it could be done can be found in the specific comments.

Another general comment, the literature review is, in my opinion, too long. Additionally, some citations were wrong. You can find suggestions on how to resolve these problems in the specific comments.

**Specific comments:**

**P.3, L.19:** To be fair, Bingen et al. (2004) retrieved three parameters of ASD based on SAGE II data. Even though the results might not be absolutely gorgeous, the work is hard to ignore.

**P.3, L.28-30:** Since here there is a discussion about the limb stratospheric aerosol products and their dependency on $P_a(\Theta)$, I think, it would be worth to mention new OSIRIS v7 product (Rieger et al., 2019). The University of Saskatchewan group minimizes the $P_a(\Theta)$ dependency by using a multiwavelength approach.

**P.3, L.34:** Generally, in my opinion, the AE discussion is unnecessarily long, it does not play any role further in the paper. I can even justify the Equation (1) and Table 1, since AE is shown on the Figures, and Table 1 gives an overview of ASD used for different products. However, the information about the AE> 2 is absolutely unnecessary and in addition, not absolutely correct. The cited paper by Schuster et al. (2006) was based on AERONET data, and its authors talked about tropospheric aerosols of different origin, where the term "small" could be applied, because there is a reference what is "small" (urban aerosol) and what is "large" (marine aerosol). However, those terms are not so easy to translate to the stratosphere, where the division to "small and large" is blurry. I suggest removing this sentence.

**P.4, Caption of table 1:** Honestly, I find (Nyaku, 2016) citation here absolutely unnecessary. The formula for Ångström exponent is given in Equation (1), and providing information which wavelengths were used for its calculation is more than sufficient.

**P.4, Table 1:** SCIAMACHY ASD parameters are wrong; those are V1.1 parameters (von Savigny et al, 2015). For V1.4 (Rieger et al., 2018), the ASD is the same as in OSIRIS retrievals. So, I suggest just to put SCIAMACHY in the same line as OSIRIS.

**P.5, L.20:** I might have missed something, but I have not found any ASD parameters retrieval information in Loughman et al. (2018).

**P.7, L.4:** Again, I might have missed something, but to define a BMLN mathematically, 6 parameters are needed, namely, 2 median radii, 2 sigmas, number density of the fine mode and number density of the coarse mode (2 number densities) or a coarse mode number density or a total number density and the CMF. In the cited paper (Malinina et al., 2018), the authors also mention 6 independent pieces of information. So, I suggest either to clear why you think 5 are enough and remove the citation, or change it to 6 data points.

**P.1, L.22 vs P.14, caption of Figure 5:** In the beginning of the manuscript, it is said that a refractive index used in this study is 1.45+0$i$, in the caption for Fig. 5 it is written that $X$ was calculated with a refractive index of 1.33. I haven't found any information in the paper, why it is so. Is there a reason for a refractive index to be 1.33? If not, in order to be consistent, I suggest replotting the $P(\Theta)$ with a refractive index of 1.45+0$i$.

**P.18, L.24:** I think, it is important to put here that in the study of Chen et al. (2018) the comparison was between collocated zonal mean profiles.

**P21. L. 30 - P.22, L.6:** I do not think it is the right conclusion which deserves 21 pages of manuscript. The authors base their conclusions on Chen et al. (2018) studies, which is fine if the conclusions were deeper. Otherwise, it shows that the study of Chen et al. (2018) was useful, no this one. To give current paper scientific significance, other conclusions should be drawn. For example, in section 3.3, the authors talk about the percentage differences in $P_a(\Theta)$. However, to understand how this percentage differences in $P_a(\Theta)$ influence limb measurements, some reference should be given. Basically, if the changes in $P_a(\Theta)$ with the "better" or "worse" fit distribution are in terms of

the UMLN within 0.01 $\mu$m change of $r_m$, then the conclusion should be, that the shape of the distribution does not play any role for the limb instruments. The $P_a(\Theta)$ changes not significantly for aerosol extinction retrievals, and everyone can use whatever distribution they want since the uncertainties in the resulting product will be the same anyway. If I understand the paper right, that is what the authors tried to say, or at least, their thoughts went this direction. This is a good and very important result; OSIRIS and SCIAMACHY teams can use their distribution, OMPS team can use theirs. There is no truth which one is right, and some other way of minimizing $P_a(\Theta)$ dependency should be found. However, this should be clearly formulated, not in a way it is done now. If the change in the phase function is comparable, for example, to the change in 0.10 $\mu$m of $r_m$ in UMLN, then I do not understand the current conclusion. The phrase "it is imperative to one to have a knowledge about the nature of the measurements from which parameters of any distribution are provided" does not deserve 22 pages of paper and should be more clearly formulated. Again, the authors write that "The overall implication of this study is to show the importance of the nature of $P_a(\Theta)$ used in the retrieval if the stratospheric aerosol extinction from limb scattering measurements", however, the limb scattering part is missing.

**Technical corrections:**

**P.2, L.26:** "..., which measures extinction directly" is not technically correct. Maybe, it is better to reformulate it to "..., which derives extinction directly" or something similar?

**P.6, Equation 2:** I don't think the multiplication signs are necessary here.

**P.6, L.17:** Maybe, it is better to write "logarithm" instead of "log".

**P.8, L.12:** I think, it is easier to put this link into references; here, the link is a bit out of place.

**Figures 3, 6, 9, 10:** There is a first "A" missing in SCIAMACHY on those Figures.

**P. 14, L. 12-13:** If you work in Latex, it would be nice to put a tilde in between 0.3 and $\mu$m.

**P. 1, L. 23-24:** It would be nice to remove space after "Station" and a bracket sign.

**P. 20, caption of Figure 10:** Maybe, it is better to put "panel on the left" and "panel on the right" instead of "figure"?

**P. 22, L. 7:** Again, maybe it is better to put the link into the references or split it somehow?

**References:**

Rieger, L. A., Zawada, D., J,Bourassa, A. E., and Degenstein, D. A. (2019). A multiwavelength retrieval approach for improved OSIRIS aerosol extinction

retrievals. Journal of Geophysical Research: Atmospheres, 124, 7286–7307. https://doi.org/10.1029/2018JD029897

von Savigny, C., Ernst, F., Rozanov, A., Hommel, R., Eichmann, K.-U., Rozanov, V., Burrows, J. P., and Thomason, L. W.: Improved stratospheric aerosol extinction profiles from SCIAMACHY: validation and sample results, Atmos. Meas. Tech., 8, 5223–5235, https://doi.org/10.5194/amt-8-5223-2015, 2015.

---

## Author Response (AR2)

**Reactions to Specific Comments from Anonymous Referee #1    Report #1**

**General comments:**

In my opinion the authors did a good job responding to my comments on the previous version. I have some addition (mainly minor) comments and ask the authors to consider them. The paper is very well written and should be accepted after a minor revision.

**Specific comments:**

**1-** Page 1, line 10: "... the gamma distribution also fits the CARMA ..."
Why "also"? I think "also" doesn't fit here.
**Response:** The word "also" has been removed from the sentence.

**2-** Page 2, line 3: "Antartic" -> "Antarctic"
**Response:** "Antartic" has been changed to "Antarctic"

**3-** Page 2, line 9: "years has" -> "years have"
**Response:** "years has" has been changed to "years have"

**4-** Page 2, line 14: "pyrocumulonumbus" -> "pyrocumulonimbus" or just "pyrocumulus"
**Response:** "pyrocumulonumbus" has been changed to "pyrocumulus"

**5-** Page 3, line 34: "AE values greater than 2 are indicative of small particles"
Without mentioning specific sizes this statement does not convey much information. What is "small"? Please be more specific or remove the statement. Or perhaps you just want to say that the AE depends on particle size and that AE decreases with increasing particle size (at least in a limited size range)?

**Response:** The sentence containing the above statement "AE values greater than 2 are indicative of small particles" has been removed.

**6-** Page 4, line 13: "when the exact AE used for the radiance simulations is applied in the extinction retrievals"
I read this sentence several times, but I'm not sure I understand what it means. Can you clarify? How is the AE used?

**Response:** Rieger et al. (2018) used an assumed particle size distribution profile to simulate radiance, and at each altitude the Ångström Exponent (calculated between 525nm and 750nm extinction) is different. In performing the extinction retrievals, the parameters of the size distribution that reproduces the same Ångström exponent as the one used in the radiance simulation is what was referred to as "the exact AE".
The sentence has been reworded to read;

"Rieger et al. (2018) have shown that when the radiance is simulated to include coarse mode particles in the atmosphere with an assumed AE, then the differences between the lognormal parameters used in the simulation and the retrieval induces errors in the retrieved aerosol extinction as function of AE which corresponds to 30% for OSIRIS geometries and 50% for SCIAMACHY geometries. This is because the phase functions of BMLN distributions vary more widely for a given AE and this leads to a complicated relationship with the retrieved error. "

More details about this technique can be found in the reference listed below:

Rieger, L. A., Malinina, E. P., Rozanov, A. V., Burrows, J. P., Bourassa, A. E., and Degenstein, D. A.: A study of the approaches used to retrieve aerosol extinction, as applied to limb observations made by OSIRIS and SCIAMACHY, Atmospheric Measurement Techniques, 11, 3433–3445, https://doi.org/10.5194/amt-11-3433-2018, https://www.atmos-meas-tech.net/11/3433/2018/, 2018.

**7-**Page 6, equation (2): The solid dots in the denominators are unusual? This should be $\times$, right?
**Response:** The solid dot have been removed from the denominator of the equation.

**8-** Page 6, line 10: "the number of usable measurements decreases" Not really clear, what this refers to? A decrease of the number of usable measurements over time, with increasing channel etc.? Please specify.

**Response:** The sentence containing the above statement has been rephrased to read "While the OPCs in use since 1991 employ 8 to 12 aerosol channels, the number of measurements decrease with increasing radius channels and altitude as the concentration of the larger particles decrease below detection thresholds."

**9-** Page 7, line 15: "radii size" sound somewhat tautological.
**Response:** "radii size" has been replaced by the phrase "according to the magnitudes of the radii"

**10-** Page 8, line 21: "number size distribution" is not a fully correct description and may be misleading.

**Response:** The phase in the sentence referenced to has been omitted and the new sentence now reads:
"A difficulty with this distribution, as stated by Wilks (2011), is that it is more tedious to work with the gamma distribution because the two parameters do not correspond exactly to the physical parameters of the size distribution of the sampled sampled data, as is the case for the lognormal distribution."

**11-** Page 9, line 1: "aerosols loads" -> "aerosol loads" ?
**Response:** "aerosols loads" has been replaced with "aerosol loads"

**12-**Page 10, line 14: "The shape of the phase functions has been observed to depend on magnitude of the median radius ($r_m$) in the case of the UMLN distribution" The shape of the phase function certainly also depends on the distribution width, right?

**Response:** The sentence "The shape of the phase functions has been observed to depend on magnitude of the median radius ($r_m$ ) in the case of the UMLN distribution and the shape parameter ($\alpha$) in the case of the gamma distribution." has been reworded to read:

"The shape of the phase functions has been observed to depend primarily on magnitude of the median radius ($r_m$) in the case of the UMLN distribution and the shape parameter ($\alpha$) in the case of the gamma distribution."

**13-** Page 12, caption Fig. 3: "SCIAMACY" -> "SCIAMACHY"
**Response:** The caption of Fig. 3: has been changed form "SCIAMACY" to "SCIAMACHY"

**14-** Page 13, line 6: Please remove parentheses after "is nearly constant for scattering angles OF"

**Response:** The parentheses have been removed.

**15-** Page 13, line 7: " Errors in estimating the number of particles within this range by the function used for fitting the data will lead to uncertainties in the phase function as shown by the $X = 1$ plot in Figure 5."

It's not fully correct to state or imply that the $X = 1$ line represents the phase function error if the small particles are neglected. The contribution of these smaller particles to the overall scattering is small because of the strong size dependence of the scattering cross section.

**Response:** The sentence "Errors in estimating the number of particles within this range by the function used for fitting the data will lead to uncertainties in the phase function as shown by the $X = 1$ plot in Figure 5." has been rewritten to read:

" Errors in estimating the number of particles within this range by the function used for fitting the data may lead to uncertainties in the phase function; however the contribution of particles within this range to the overall scattering is small due to the strong size dependence of the scattering cross section. Compare the phase function plots shown by the $X = 1$ and $X = 3$ lines in Figure 5."

**16-** Page 14, Figure 5: Something is wrong here. The case with $X = 10$ should have the most pronounced forward scattering peak, i.e. a larger value of the phase function at a scattering angle of 0 deg compared to the cases. Please check.

**Response:** The case with $X = 10$ has the most pronounced peak at a scattering angle of $0°$. The figure 5 shown previously had scattering angles ranging for $30°$ to $130°$ and using a refractive index of $1.33 + 0i$. The current figure has being computed using a refractive index of $1.45 + 0i$ and the scattering angle ranges from $20°$ to $130°$ ( scattering angles observed by three limb scattering instruments mentioned in this study).

**17-** Page 15, line 4: "compered" -> "compared"
**Response:** "compered" has been changed to "compared"

**18-** Page 16, caption Fig. 6: "SCIAMACY" -> "SCIAMACHY"
**Response:** "SCIAMACY" has been changed to "SCIAMACHY"

**19-** Page 17, line 14: "and degassing volcanoes that are not explosive in nature"

What about the eruptions of Kasatochi, Sarychev Peak, Nabro, Calbuco etc. that occurred during this period of time? They were not particularly strong, but they were explosive, injected sulfur compounds into the stratosphere and affected the aerosol size distribution. If this eruptions were not considered, how can the size distribution be modelled accurately?

**Response:** For this study, we were more interested in the background aerosol which reflected the anthropogenic and the non-volcanic sources. Also the volcanoes mentioned above, although explosive in nature did not reach the altitudes (20 -25km) at the location that was considered in this study.

**20-** Page 17, line 18: "distributions fits" -> "distribution fits"
**Response:** "distributions fits" has been replaced with "distribution fits"

**21-** Page 17, line 22: "are computed" -> "is computed"

**Response:** "are computed" has been replaced with "is computed"

**22-** Page 17, line 22: Same sentence: Have you tested how the $chi^2$ results change if only the bins are considered which are used for the fitting process? The results are probably different, but would be more consistent with the results obtained when fitting the OPC measurements. I suggest checking.

**Response:** The $chi^2$ results have been tested when they are computed using only the bins used in the fits. The results show a decrease in the $chi^2$ values and an increase in the percentile values, and this results are not different from those obtained when the $chi^2$ are computed to include the excluded bins during the fitting process. See figures below.

[Figure]

**Figure 1.** Unimodal lognormal and gamma distribution fits to the normalized CARMA model data. The blue data points are excluded during the fitting procedure and the validation of the fits. The green lines are the gamma distribution fits and the purple lines are the UMLN distribution fits.

[Figure]

**Figure 2.** Percentile values computed from the updated minimized $\chi^2$ of the fits of both the UMLN and gamma distributions to determine the level of confidence for which either distribution is chosen to describe the CARMA model data.

**23-** Page 18, line 17: "The good comparison ... provides evidence for the agreement" sounds tautological.

**Response:** The above sentence has been revised to read:
"The good comparison shown by the phase functions derived from the CARMA model and those from the OPC dataset at Laramie, Wyoming, provides evidence that the CARMA model outputs agree with the Wyoming OPC measurements. This provides a justification for the use of the CARMA model results at other locations on the Earth and for periods with moderate volcanic activity."

**24-** Page 18, line 23: "Space Station )" -> "Space Station)"
**Response:** The white space at the end of Space Station has been removed.

**25-** Page 21, line 6: "the scattered radiance arises from the aerosol phase function"
This is not well phrased, I think. The radiance arises (in part) from scattering by aerosols - whose angular distribution is described by the phase function ... But the scattered radiance does not arise from the phase function.

**Response:** The above sentence has been updated to read:
"Along the LOS of the sensor, the scattered radiance arises (in part) from the scattering by aerosols, whose angular distribution is described by the phase function"

**26-** Page 21, line 22: "some what" -> "somewhat"
**Response:** "some what" has been replaced with "somewhat"

**27-** Page 21, line 22: "because of its insensitivity to particles between"
Is this really the case? What kind of "insensitivity" do you mean? I don't really understand this statement.
**Response:** The phrase "because of its insensitivity" has been taken out of the sentence.

**Reactions to Specific Comments from Anonymous Referee #2    Report #3**

**General comments:**

In this paper, the authors study which aerosol particle size (ASD) distribution fits better OPC measurements and/or CARMA model output. Here, two ASDs were taken into consideration, namely, unimodal lognormal (UMLN) and gamma distribution. The authors also look at how the aerosol phase function ($P_a(\Theta)$) changes with different ASDs and particle size range taken into consideration. While, in general, this revision of the manuscript got significant improvements in comparison to the previous version, the authors still did not clarify all the points from the previous review. Because of that, I think, another revision of the manuscript should be done. Even though the conclusions were reformulated, they are not yet sufficient to be published. If I understood this paper correctly, its take-home message is "you should know where your ASD came from". This message does not need 21 pages of studies. Reading this revision of the manuscript, I got an impression that the authors tried to say that UMLN is a good fit for OPC measurements, while CARMA model output is better fitted with gamma distribution. However, both of the distributions are correct depending on the source of information one uses, and there is no truth, which of them should be used in limb retrievals. This is a very important result for the limb community, which looks for the last 15 years for the solution of the aerosol extinction coefficient dependency on $P_a(\Theta)$ problem, and in particular, tries different ASDs in order to improve products. If I am correct in my understanding, then the conclusions should be reformulated this way, providing numerical proof. The authors quantify the changes in the $P_a(\Theta)$; however, they do not explain how this change in $P_a(\Theta)$ influences the resulting limb extinctions, which in my opinion is absolutely essential. Again, if the authors do not want to deal with limb retrievals, that is fine, but then the paper should be revised to remove a long introduction about limb instruments. If the authors want to leave the discussion on how this study is important for the limb community, then they should show it differently. The suggestions on how it could be done can be found in the specific comments. Another general comment, the literature review is, in my opinion, too long. Additionally, some citations were wrong. You can find suggestions on how to resolve these problems in the specific comments.

**Response:** The goal of this paper is to show that aerosol phase functions derived from the same data can be different depending on the size distribution used to fit the data. We showed the effect of using two single mode distributions (UNLN and gamma) to fit OPC data which had at least one measurement between 0.01 μm - 0.1 μm and when there was no measurement within that range. Our results indicated that when there was at least a measurement within that radii range, the UMLN distribution was a good fit to the data. Subsequently, when there was no measurement bin within that range, the gamma distribution was the better fit since the derived phase functions from this distribution parameters were similar as compared to those of the UMLN. The study was further extended to use the CARMA model outputs which has three data points within the radii range 0.01 μm - 0.1 μm. Here results indicated that gamma distribution was a better fit to this model output. Because, the main objective of this study was not to perform extinction retrievals, we referred the reader to a parallel study (Chen et al. 2018), who made a comparison between collocated zonal mean profiles derived using a gamma distribution and a lognormal distributon. The introduction about limb instrument is necessary because the aerosol phase function is primarily used with limb scatter measurements.

I don't think the literature review is too long.

**Specific comments:**

**1-** P.3, L.19: To be fair, Bingen et al. (2004) retrieved three parameters of ASD based on SAGE II data. Even though the results might not be absolutely gorgeous, the work is hard to ignore.
**Response:** The work of Bingen et al. (2004) has been mentioned and referenced in the script.

**2-** P.3, L.28-30: Since here there is a discussion about the limb stratospheric aerosol products and their dependency on $P_a(\Theta)$, I think, it would be worth to mention new OSIRIS v7 product (Rieger et al., 2019). The University of Saskatchewan group minimizes the $P_a(\Theta)$ dependency by using a multiwavelength approach.

**Response:** OSIRIS v7 product (Rieger et al., 2019) has been mentioned.

**3-** P.3, L.34: Generally, in my opinion, the AE discussion is unnecessarily long, it does not play any role further in the paper. I can even justify the Equation (1) and Table 1, since AE is shown on the Figures, and Table 1 gives an overview of ASD used for different products. However, the information about the AE> 2 is absolutely unnecessary and in addition, not absolutely correct. The cited paper by Schuster et al. (2006) was based on AERONET data, and its authors talked about tropospheric aerosols of different origin, where the term "small" could be applied, because there is a reference what is "small" (urban aerosol) and what is "large" (marine aerosol). However, those terms are not so easy to translate to the stratosphere, where the division to "small and large" is blurry. I suggest removing this sentence.

**Response:** Equation (1) is necessary to show the reader that the Ångström exponent is calculated from the extinction profiles of two wavelengths. Also the sentence "AE values greater than 2 are indicative of small particles (Schuster et al., 2006)" has been removed.

**4-** P.4, Caption of table 1: Honestly, I find (Nyaku, 2016) citation here absolutely unnecessary. The formula for Ångtröm exponent is given in Equation (1), and providing information which wavelengths were used for its calculation is more than sufficient.

**Response:** (Nyaku, 2016) citation has been taken away.

**5-** P.4, Table 1: SCIAMACHY ASD parameters are wrong; those are V1.1 parameters (von Savigny et al, 2015). For V1.4 (Rieger et al., 2018), the ASD is the same as in OSIRIS retrievals. So, I suggest just to put SCIAMACHY in the same line as OSIRIS.

**Response:** The SCIAMACHY ASD parameters for the two versions have been updated.

**6-** P.5, L.20: I might have missed something, but I have not found any ASD parameters retrieval information in Loughman et al. (2018).

**Response:** "Loughman et al. (2018)" has been removed from the list of citations in this line.

**7-** P.7, L.4: Again, I might have missed something, but to define a BMLN mathematically, 6 parameters are needed, namely, 2 median radii, 2 sigmas, number density of the fine mode and number density of the coarse mode (2 number densities) or a coarse mode number density or a total number density and the CMF. In the cited paper (Malinina et al., 2018), the authors also mention 6 independent pieces of information. So, I suggest either to clear why you think 5 are enough and remove the citation, or change it to 6 data points.

**Response:** I agree with you that a BMLN function would need 6 parameters to fit it, namely, 2 median radii, 2 sigma values and 2 number density values ( fine mode and coarse mode). This means that a minimum of 6 equations with 6 unknowns. For our fits the data is first normalized, so that the total number density ($N = N_{cm} + N_{fm} = 1$). Then by the definition of CMF ( which is the ratio of the course mode number density to the total number density) it $CMF = N_{cm}$ and $(1 - CMF) = N_{fm}$. Thus replacing the number density of the fine mode ($N_{fm}$) by $1 - CMF$, the BMLN function is reduced from 6 parameters to 5. The cited paper (Malinina et al., 2018) has been removed.

**8-** P.1, L.22 vs P.14, caption of Figure 5: In the beginning of the manuscript, it is said that a refractive index used in this study is $1.45 + 0i$, in the caption for Fig. 5 it is written that X was calculated with a refractive index of $1.33$. I haven't found any information in the paper, why it is so. Is there a reason for a refractive index to be $1.33$? If not, in order to be consistent, I suggest replotting the $P(\theta)$ with a refractive index of $1.45 + 0i$.

**Response:** The phase functions have been replotted using a refractive index of $1.45 + 0i$.

**9-** P.18, L.24: I think, it is important to put here that in the study of Chen et al. (2018) the comparison was between collocated zonal mean profiles.
**Response:** The phrase "zonal mean profiles" has been added to the sentence.

**10-** P21. L. 30 - P.22, L.6: I do not think it is the right conclusion which deserves 21 pages of manuscript. The authors base their conclusions on Chen et al. (2018) studies, which is fine if the conclusions were deeper. Otherwise, it shows that the study of Chen et al. (2018) was useful, no this one. To give current paper scientific significance, other conclusions should be drawn. For example, in section 3.3, the authors talk about the percentage differences in $P_a(\Theta)$. However, to understand how this percentage differences in $P_a(\Theta)$ influence limb measurements, some reference should be given. Basically, if the changes in $P_a(\Theta)$ with the "better" or "worse" fit distribution are in terms of the UMLN within 0.01μm change of $r_m$, then the conclusion should be, that the shape of the distribution does not play any role for the limb instruments. $P_a(\Theta)$ changes not significantly for aerosol extinction retrievals, and everyone can use whatever distribution they want since the uncertainties in the resulting product will be the same anyway. If I understand the paper right, that is what the authors tried to say, or at least, their thoughts went this direction. This is a good and very important result; OSIRIS and SCIAMACHY teams can use their distribution, OMPS team can use theirs. There is no truth which one is right, and some other way of minimizing $P_a(\Theta)$ dependency should be found. However, this should be clearly formulated, not in a way it is done now. If the change in the phase function is comparable, for example, to the change in 0.10 μm of $r_m$ in UMLN, then I do not understand the current conclusion. The phrase "it is imperative to one to have a knowledge about the nature of the measurements from which parameters of any distribution are provided" does not deserve 22 pages of paper and should be more clearly formulated. Again, the authors write that "The overall implication of this study is to show the importance of the nature of $P_a(\Theta)$ used in the retrieval if the stratospheric aerosol extinction from limb scattering measurements", however, the limb scattering part is missing.

**Response:** This paper, as the title indicates seeks to compare the phase functions derived from lognormal and gamma distribution fits to OPC data at two altitudes 20 km and 25 km, and then extend this comparison to model outputs from CARMA. Also, because this study was done in parallel with the Chen et al. (2018) studies, we found it fit to base some of our conclusion (especially the extinction retrievals) on this study.

In section 3.3, UMLN and gamma distribution fits are made to the CARMA microphysical model results at Wyoming and phase functions are computed from the parameters of the fits. The derived phase functions are then compared with the phase functions obtained from OPC in the form of percentage difference to show and to quantify the variation that was seen. Since, this paper did not seek to perform extinction retrievals, the authors referred the readers to the Chen et al. (2018) study which showed an improvement in the extinction retrievals when the gamma distribution is used as the aerosol size distribution.

This paper intends to show that OSIRIS, SCIAMACHY and OMPS versions 0.5 and 1.0 all use a lognormal aerosol size distribution that was based on fits made to OPC data. Most of the OPC data did not have a measurement between 0.01 μm ad 0.1 μm. Using CARMA model outputs which had at least a measurement between this particle size range and fitting the two unimodal distributions (lognormal & gamma) to it was to determine which of these two distribution predicted the outputs

within this radii range . The parameters of each distribution are then used to compute the phase function and the percentage difference between these phase functions are then compared.

The authors make is assertion "The overall implication of this study is to show the importance of the nature of $P_a(\Theta)$ used in the retrieval if the stratospheric aerosol extinction from limb scattering measurements", of the importance of the phase function in the limb view geometry when performing extinction retrievals. Also because we made mention of the parallel study Chen et al. (2018), for which the retrievals are made from limb scattering measurements, it is imperative to include that phase in the sentence.

**Technical corrections:**

**1-** P.2, L.26: "..., which measures extinction directly" is not technically correct. Maybe, it is better to reformulate it to "..., which derives extinction directly" or something similar?
**Response:** The phase "which measures extinction directly" has been replaced with "that derives extinction directly".

**2-** P.6, Equation 2: I don't think the multiplication signs are necessary here.
**Response:** The multiplication signs have been removed from the equation.

**3-** P.6, L.17: Maybe, it is better to write "logarithm" instead of "log".
**Response:**  "log" in this sentence has been replaced by "logarithm".

**4-** P.8, L.12: I think, it is easier to put this link into references; here, the link is a bit out of place.
**Response:** The line has been put into references.

**5-** Figures 3, 6, 9, 10: There is a first "A" missing in SCIAMACHY on those Figures.
**Response:** The spelling of " SCIAMACHY" has been corrected.

**6-** P. 14, L. 12-13: If you work in Latex, it would be nice to put a tilde in between 0.3 and µm.
**Response:** This has been noted.

**7-** P. 1, L. 23-24: It would be nice to remove space after "Station" and a bracket sign.
**Response:** The white space has been removed.

**8-**P. 20, caption of Figure 10: Maybe, it is better to put "panel on the left" and "panel on the right" instead of "figure"?
**Response:** The word "figure" in the caption of Figure 10 has been replaced by "panel".

**9-** P. 22, L. 7: Again, maybe it is better to put the link into the references or split it somehow?
**Response:**  The line has been put into references.

**Relevant Change**

The following change has been made in the manuscript:

– The Mie phase functions have now been derived with a refractive index of $1.45 + 0i$ instead of $1.33 + 0i$.

[Figure]

. The figure on the left hand side derived with a refractive index of $1.33 + 0i$ has been updated to the one on the right hand side (refractive index is $1.45 + 0i$).

[revised manuscript text omitted]

---

## Author Response (AR3)

**Reactions to Associate Editor Minor Revision:**

Associate Editor Decision: Publish subject to minor revisions (review by editor) (20 Jan 2020) by Troy Thornberry

**Comments to the Author:**

I think overall the manuscript is in good shape for publication, although the conclusion/summary section perhaps needs some attention as was noted by a reviewer last fall.

The conclusions/summary should include more quantitative results, at least to the level of the abstract (if not more). The discussion of the fit comparison in paragraph 2 ( $\chi^2$  details) is wordy and repetitive. Worth noting again that the particle size measurements are expressed as radii. What would be good to have (as noted by a reviewer previously) would be some estimate of the importance (significance) of the Pa( $\theta$ ) measurement/model difference (Fig 10) on LS retrievals—doesn't need to be extensive, but it would seem to be what is needed to assess the overall impact of the work.

**Response:**

1- Some additional quantitative results have been included in the conclusions/summary.

2- The discussion of the fit comparison in paragraph 2 of the conclusion/ summary has been revised to now read:

"We have investigated fitting a unimodal lognormal (UMLN) and a gamma distribution to the 2008 to 2017 Wyoming in situ LPC measurements, which include a measurement below 0.1  $\mu$ m, for altitudes 20 km and 25 km. The parameters of the distributions were found by minimizing the  $\chi^2$  test statistic between the measurements and the theoretical distributions. As a first step, we assumed that the stratospheric aerosol is distributed with a single mode during the background conditions and could be fitted to either of the two distributions. Typically, both cumulative distributions are found to be good representatives for these measurements as was suggested by the  $\chi^2$  values. To discriminate between them, a  $\chi^2$  goodness of fit test applied showed that to a 10% level of confidence the UMLN was the better of the two distributions as it fitted all data at the two altitudes and for all the months of data that were considered."

**3-** This study was specifically made not to perform aerosol extinction retrievals. As a result, readers are referred to a parallel study (Chen et al. 2018), where a comparison was made between collocated zonal mean aerosol profiles derived using a gamma distribution whose parameters were derived from the CARMA model outputs and a lognormal distribution whose parameters were taken from OPC measurements.

From that study, the significance of the  $P_a(\Theta)$  on LS retrievals was shown by perturbing each the gamma distribution parameters used ( $\alpha = 1.8$  and  $\beta = 20.5$ ) and then studying the effect on the retrieved aerosol extinction for the range of scattering angles viewed during a single OMPS/LP orbit. The results showed that a  $\pm 10\%$  change in the gamma distribution parameter  $\beta$  would produce a  $\pm 10\%$  change in the calculated  $P_a(\Theta)$  at scattering angles between 70° and 100°, whereas a  $\pm 10\%$  change in  $\alpha$  results in a  $\pm 3\%$  change in  $P_a(\Theta)$  for  $\Theta > 70^\circ$ .

The changes in the retrieved aerosol extinction as shown in Figure 3 of Chen et al. (2018), were found to be approximately anti-correlated with the phase function variation. That is, the fractional change in the retrieved aerosol extinction was about half of the change in the  $P_a(\theta)$  depending on the single scattering angle. This showed that underestimating the  $P_a(\Theta)$  would

overestimate the retrieved aerosol extinction and vice-versa.

Chen, Z., Bhartia, P. K., Loughman, R., Colarco, P., and DeLand, M.: Improvement of stratospheric aerosol extinction retrieval from OMPS/LP using a new aerosol model, Atmospheric Measurement Techniques, 11, 6495–6509, 2018.

**A few specific comments to consider, beyond what will likely be addressed in copy editing:**

**1-** P2: SO2 doesn't need to be italicized. **Response:** This has been noted and updated.

**2-** P2,L7: "SO2 is also injected"—the "also" doesn't have a referent since you have not mentioned other SO2 pathways to the stratosphere.

Response: "also" has been removed from the sentence. The sentence now reads:

"Through large volcanic eruptions,  $SO_2$  is injected directly into the stratosphere leading to an increased aerosol concentration that lasts for several years as was observed after the eruptions of El Chichón (Mexico, 1982) and Pinatubo (Philippines, 1991)."

**3-** P2,L7-13: the volcanic role in stratospheric aerosol is missing nuance—blanket statements about injection height and duration of impact are simplistic. Should at least express as ranges: "effects can last up to several years", "injections can reach", might mention impact dependence on latitude

**Response:** Lines 7 - 13 has been rewritten to now read:

Large volcanic eruptions directly inject plumes of  $SO_2$  into the atmosphere that can reach beyond the tropical tropopause and into the stratosphere, where the  $SO_2$  is oxidized and increases the load of the stratospheric aerosol particles (Solomon et al., 2011). This effect can last up to several years as was observed after the eruptions of El Chichón (Mexico, 1982) and Pinatubo (Philippines, 1991). The past 20 years have not experienced any large volcanic eruptions, but during this period the stratospheric aerosol load has been controlled by "moderate" but recurring volcanic eruptions that have been reported to be a primary source of the enhancement of global aerosol content (Vernier et al., 2011b; Neely III et al., 2013; Mills et al., 2016; Berthet et al., 2017). These moderate volcanic plumes can reach between 18-20 km in the lower stratophere, and through the upwelling branch of the Brewer-Dobson circulation, they are lofted into the mid-stratosphere up to 25 km altitude in about one year (Vernier et al., 2011b). Also, moderate volcanic eruptions like the Sarychev eruption in 2009 located in high-latitudes can enhance the aerosol load in the tropical stratosphere and even impact the other hemisphere (Wu et al., 2017).

**4-** Good additional references for the role of moderate volcanic activity in the stratospheric aerosol layer would be: R. R. Neely III, O. B. Toon, S. Solomon, J. P. Vernier, C. Alvarez, J. M. English, K. H. Rosenlof, M. J. Mills, C. G. Bardeen, J. S. Daniel, J. P. Thayer, Recent anthropogenic increases in SO2 from Asia have minimal impact on stratospheric aerosol Geophysical Research Letters, doi:10.1002/grl.50263, 2013. **Response:** This reference has been cited.

5- Mills, M. J., et al. (2016), Global volcanic aerosol properties derived from emissions, 1990–2014, using CESM1(WACCM),
J. Geophys. Res. Atmos., 121, doi:10.1002/2015JD024290.
Response: This reference has been cited.

**6-** P2:L14: "pyrocumulus clouds from large wildfires that can inject" **Response:** This has been noted and the senctence has been updated.

**7-** P2:L15: see Neeley et al ref on ASM SO2 source. There are recent Pengfei Yu et al and Felix Ploeger et al papers that would be useful to reference for the ASM statement. **Response:** The following citations have been added

Vernier, J.-P., Thomason, L., and Kar, J.: CALIPSO detection of an Asian tropopause aerosol layer, Geophysical Research Letters, 38, 2011a.

Yu, P., Rosenlof, K. H., Liu, S., Telg, H., Thornberry, T. D., Rollins, A. W., Portmann, R. W., Bai, Z., Ray, E. A., Duan, Y., et al.: Efficient transport of tropospheric aerosol into the stratosphere via the Asian summer monsoon anticyclone, Proceedings of the National Academy of Sciences, 114, 6972–6977, 2017.

Ploeger, F., Konopka, P., Walker, K., and Riese, M.: Quantifying pollution transport from the Asian monsoon anticyclone into the lower stratosphere, Atmospheric Chemistry and Physics, 17, 7055, 2017.

**8-** P2:L22: RI of 1.45+0i is assumed for all wavelengths? **Response:** Yes. The sentence has been revised to read:

"In this study, a refractive index of 1.45 + 0i is assumed as appropriate for hydrated sulfuric acid (Palmer and Williams, 1975) and is used for all wavelenghs."

**9-** P3,L34: "retrieval" should be "retrievals" **Response:** This has been noted and updated.

**10-** P5,L24: "aerosol concentration measurements" **Response:** This has been noted and updated.

**11-** P5,L27: "sulfuric acid vapor"?

**Response:** The sentence has been revised to read:

"The nucleation mode is associated with new particle formation from sulfuric acid vapor which quickly coagulate to form larger particles (Hamill et al., 1997), and the accumulation mode associated with particle growth by condensation of the vapor on the existing particles (Steele and Turco, 1997). "

**12-** P6,L4: calibration is used in addition to Mie theory to convert signal to particle size, correct? **Response:** The sentence has been revised to read:

"The instrument caliberation and the instrument response function which depends on the light source, the detector efficiency, and Mie theory are used to determine aerosol size from the intensity of the scattered light." **13-** P21,L7: suggest "molecular scattering and trace species absorption" **Response:** The sentence has been revised to read: "Along the LOS of the sensor, the scattered radiance arises (in part) from the scattering by aerosols, whose angular distribution is described by the phase function,  $P_a(\Theta)$ , but is attenuated by molecular scattering and trace species absorption, making untangling of the information content in these measurements very complicated."

**A comparison of lognormal and gamma size distributions for characterizing the stratospheric aerosol phase function from OPC measurements**

Ernest Nyaku1, Robert Loughman1, Pawan K. Bhartia2, Terry Deshler3, Zhong Chen4, and Peter R. Colarco2

1Center of Atmospheric Science, Hampton University, Hampton
 2NASA Goddard Space Flight Center, Greenbelt, Maryland, 20771, USA
 3Department of Atmospheric Science, University of Wyoming, Laramie, Wyoming
 4Science Systems and Applications, Inc. (SSAI), 10210 Greenbelt Road, Suite 600, Lanham, Maryland 20706, USA

Correspondence: Ernest Nyaku (ernest.nyaku@hamptonu.edu)

**Abstract.**

[revised manuscript text omitted]